# Divergent Cl$^-$ and H$^+$ pathways underlie transport coupling and gating in CLC exchangers and channels

Lilia Leisle[1†‡], Yanyan Xu[2,3†§], Eva Fortea[4], Sangyun Lee[1], Jason D Galpin[5], Malvin Vien[1], Christopher A Ahern[5], Alessio Accardi[1,4,6*], Simon Bernèche[2,3*]

[1]Department of Anesthesiology, Weill Cornell Medical College, New York, United States; [2]SIB Swiss Institute of Bioinformatics, University of Basel, Basel, Switzerland; [3]Biozentrum, University of Basel, Basel, Switzerland; [4]Department of Physiology and Biophysics, Weill Cornell Medical College, New York, United States; [5]Department of Molecular Physiology and Biophysics, University of Iowa Carver College of Medicine, Iowa City, United States; [6]Department of Biochemistry, Weill Cornell Medical College, New York, United States

*For correspondence:
ala2022@med.cornell.edu (AA);
simon.berneche@me.com (SB)

[†]These authors contributed equally to this work

Present address: [‡]Institute of Physiology, RWTH Aachen University, Aachen, Germany; [§]Tan Kah Kee College, Xiamen University, Zhangzhou, Fujian, China

Competing interests: The authors declare that no competing interests exist.

**Abstract** The CLC family comprises H$^+$-coupled exchangers and Cl$^-$ channels, and mutations causing their dysfunction lead to genetic disorders. The CLC exchangers, unlike canonical 'ping-pong' antiporters, simultaneously bind and translocate substrates through partially congruent pathways. How ions of opposite charge bypass each other while moving through a shared pathway remains unknown. Here, we use MD simulations, biochemical and electrophysiological measurements to identify two conserved phenylalanine residues that form an aromatic pathway whose dynamic rearrangements enable H$^+$ movement outside the Cl$^-$ pore. These residues are important for H$^+$ transport and voltage-dependent gating in the CLC exchangers. The aromatic pathway residues are evolutionarily conserved in CLC channels where their electrostatic properties and conformational flexibility determine gating. We propose that Cl$^-$ and H$^+$ move through physically distinct and evolutionarily conserved routes through the CLC channels and transporters and suggest a unifying mechanism that describes the gating mechanism of both CLC subtypes.

## Introduction

The CLC (ChLoride Channel) family is comprised of Cl$^-$ channels and H$^+$-coupled exchangers whose primary physiological task is to mediate anion transport across biological membranes (*Accardi, 2015*; *Jentsch and Pusch, 2018*). The human genome encodes nine CLC homologues, four (CLC-1,–2, -Ka, -Kb) are Cl$^-$ channels that reside in the plasma membrane and five (CLC-3 through −7) are 2 Cl$^-$:1 H$^+$ antiporters that localize to intracellular compartments along the endo-lysosomal pathway. Mutations in at least five of the human CLC genes lead to genetically inherited disorders of muscle (Thomsen and Becker type *Myotonia congenita*), kidney (Bartter Syndrome Type III and IV, Dent's disease), bone (Osteopetrosis) and central nervous system (neuronal ceroid lipofuscinosis, retinal degeneration, deafness, syndromic intellectual disability, seizure disorders), highlighting the fundamental roles of CLC channels and transporters in human physiology (*Jentsch, 2008*; *Stauber et al., 2012*; *Hu et al., 2016*; *Jentsch and Pusch, 2018*; *Palmer et al., 2018*). Several disease-causing mutations occurring in CLC channels and transporters impair the response to the physiological stimuli regulating their activity, such as voltage, pH and nucleotide concentration (*Accardi, 2015*; *Alekov, 2015*; *Bignon et al., 2018*; *Jentsch and Pusch, 2018*).

High-resolution structural information on the CLC-ec1 and cmCLC exchangers (*Dutzler et al., 2002*; *Feng et al., 2010*) as well as CLC-K and CLC-1 channels (*Park et al., 2017*; *Park and*

*MacKinnon, 2018*; *Wang et al., 2019*) revealed that both CLC subtypes share a common dimeric architecture, where each monomer forms physically distinct and functionally independent ion permeation pathways. This pathway is defined by three anionic binding sites (*Dutzler et al., 2002*; *Dutzler et al., 2003*; *Accardi and Picollo, 2010*; *Feng et al., 2010*; *Park et al., 2017*; *Figure 1A*). The external and central sites, $S_{ext}$ and $S_{cen}$, are alternatively occupied by the permeant anions or by the negatively charged side chain of a conserved glutamic acid, $Glu_{ex}$ (*Figure 1A*). The internal site, $S_{int}$, binds anions weakly and likely serves as a recruitment site of intracellular permeating ions (*Lobet and Dutzler, 2006*; *Picollo et al., 2009*). The presence of a fourth binding site, $S_{ext*}$, in direct contact with the extracellular solution has been proposed based on electrostatic calculations and is supported by electrophysiological (*Lin and Chen, 2000*), structural (*Park et al., 2019*), and molecular dynamics simulation studies (*Faraldo-Gómez and Roux, 2004*; *Mayes et al., 2018*).

The CLC Cl⁻ pore can adopt at least three conformations, differentiated by the position and protonation state of $Glu_{ex}$ (*Dutzler et al., 2002*; *Dutzler et al., 2003*; *Feng et al., 2010*; *Figure 1A*). Extensive functional work suggests that cycling through these conformations underlies the Cl⁻/H⁺ exchange cycle in the CLC transporters and gating in the CLC channels (*Lísal and Maduke, 2008*; *Feng et al., 2010*; *Feng et al., 2012*; *Basilio et al., 2014*; *Khantwal et al., 2016*; *Vien et al., 2017*). All proposals accounting for the CLC exchange cycle suggest that these transporters do not utilize a classical 'ping-pong' mechanism of antiport, where the transporter sequentially interacts with one substrate at a time (*Picollo et al., 2012*). Rather, the CLCs were proposed to simultaneously bind (*Picollo et al., 2012*) and translocate Cl⁻ and H⁺ through partially congruent pathways (*Accardi et al., 2005*; *Zdebik et al., 2008*; *Figure 1B*). The H⁺ pathway is delimited by two glutamic acids that are conserved in the CLC transporters: $Glu_{in}$ serves as the intracellular proton acceptor and is distal from the Cl⁻ permeation pathway, while $Glu_{ex}$ is the extracellular intersection between the H⁺ and Cl⁻ pores (*Figure 1B*; *Accardi et al., 2005*; *Zdebik et al., 2008*). Several models have been proposed for the exchange cycle of the CLCs (*Miller and Nguitragool, 2009*; *Feng et al., 2010*; *Basilio et al., 2014*; *Khantwal et al., 2016*). However, none provided a molecular mechanism describing how the Cl⁻ and H⁺ ions bypass each other while moving in opposite directions through the permeation pathway. These proposals share the critical assumption that protonation of $Glu_{ex}$ within the pathway destabilizes its binding to $S_{cen}$ and/or $S_{ext}$, favoring its exit from the pathway. While this mechanism readily explains outward H⁺ transfer, it requires that during H⁺ influx a protonated and neutral $Glu^0_{ex}$ outcompetes the negatively charged Cl⁻ ions bound to the anion-selective $S_{ext}$ and $S_{cen}$ sites. This is an energetically unfavorable transition, which should result in intrinsic rectification of transport. Indeed, free-energy calculations show that a protonated $Glu_{ex}$ encounters a high energy barrier to enter the Cl⁻ permeation pathway (*Kuang et al., 2007*). In contrast, the CLC exchangers function with comparable efficiency in the forward and reverse directions (*Matulef and Maduke, 2005*; *Leisle et al., 2011*; *De Stefano et al., 2013*), suggesting that the entry and exit of

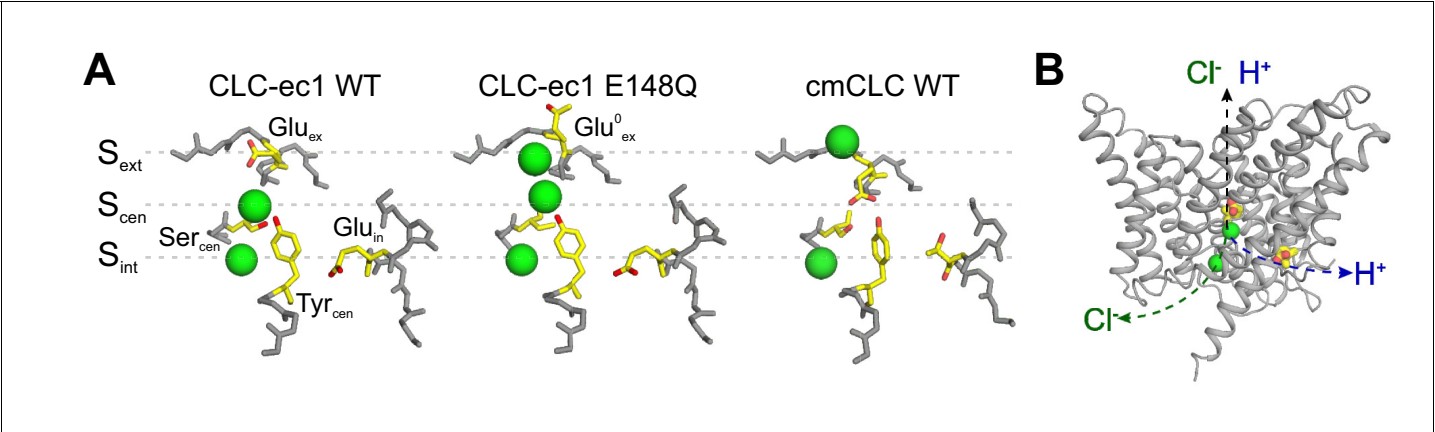

**Figure 1.** The anion pathway of the CLC Cl⁻/H⁺ exchangers. (**A**) Close up view of the Cl⁻ permeation pathway in three configurations: *left*, $Glu_{ex}$ bound to $S_{ext}$ (CLC-ec1 WT; PDBID: 1OTS); *center*, protonated $Glu_{ex}$ (mimicked by E148Q mutation) reaching out of the ion pathway (PDBID: 1OTU); *right*, $Glu_{ex}$ bound to $S_{cen}$ (cmCLC WT, PDBID: 3ORG). Selected residues are shown as sticks and Cl⁻ ions as green spheres. (**B**) The partially congruent Cl⁻ and H⁺ pathways are shown in CLC-ec1 WT structure ($Glu_{ex}$ and $Glu_{in}$ are shown in yellow, Cl⁻ ions as green spheres).

the protonated $Glu_{ex}$ from the $Cl^-$ pathway are equally favorable. Another free-energy calculation study shows that the reaction rates of deprotonation of $Glu^0_{ex}$ in both directions of the $H^+$ pathway are comparable to each other, while both reactions are coupled to $Cl^-$ at $S_{cen}$ (*Lee et al., 2016*). Further, the mechanisms regulating the release of ions from $S_{ext}$ and $S_{cen}$ and their coupling to the movement of $Glu_{ex}$ and of $H^+$ through the protein are unknown. While biochemical evidence suggests that ion release is rate-limited by a conformational step (*Picollo et al., 2009*), no release mechanism has been identified. Therefore, two key mechanistic features at the heart of the $H^+{:}Cl^-$ exchange mechanism of the CLCs, the pathways and the coupling mechanism of the substrates, remain unknown.

Here we combined molecular dynamics simulations with biochemical and electrophysiological measurements, and atomic mutagenesis to investigate the mechanism of $H^+/Cl^-$ exchange. We find that, contrary to previously proposed models, a protonated $Glu_{ex}$ does not move through the $Cl^-$ pore. Rather, we identify two highly conserved phenylalanine residues that form an aromatic slide which allows the protonated (neutral) $Glu_{ex}$ to move to and from $S_{cen}$ without directly competing with $Cl^-$ ions for passage through the anion-selective pathway. Further, we show that the rotational movement of the central phenylalanine residue, that enables the formation of the aromatic slide, is an important regulator of ion movement within the pathway, providing the molecular mechanism for the coupled exchange of $H^+$ and $Cl^-$ by the CLC transporters. Mutating these residues in prokaryotic and mammalian CLC exchangers severely impairs transport indicating that the role of these aromatic side chains is evolutionarily conserved. Since these phenylalanine residues are highly conserved throughout the CLC family, we hypothesized the role of the aromatic slide might be evolutionary conserved also between CLC channels and exchangers. Indeed, we found that they play important roles in the gating of the prototypical CLC-0 channel. Using atomic-scale mutagenesis, we probed how the aromatic slide residues interact with $Glu_{ex}$ in CLC-0, and found that the central phenylalanine interacts electrostatically with the gating glutamate, and that its conformational rotation is necessary for channel gating. We propose a novel mechanism for CLC mediated $H^+{:}Cl^-$ exchange, where the $Cl^-$ and $H^+$ pathways are distinct and intersect only near the central ion binding site.

## Results

### Molecular dynamic simulations suggest F357 controls entry and release of $Cl^-$ from $S_{cen/title}$

We used molecular dynamics simulations to probe the energetic landscape of ion movement through the permeation pathway of CLC-ec1 to ask whether it is regulated by conformational rearrangements of the pore. The first state we considered is one with $Glu_{ex}$ protonated and out of the pathway, an E148Q-like state (*Dutzler et al., 2003*; *Figure 1A*), so that all binding sites are accessible to ions. The first $Cl^-$ ion to reach the pore preferably binds to $S_{cen}$ or $S_{ext}$, this remains true when another ion is bound to $S_{extx002A}$* (*Figure 2A*, dashed line (i), 2B). When a $Cl^-$ occupies $S_{int}$, the second ion can occupy $S_{cen}$ or $S_{ext}$, though binding to $S_{cen}$ is less favorable by 2 kCal $mol^{-1}$ (*Figure 2A*, dashed line (ii), 2B). Unexpectedly, our calculations suggest that simultaneous binding of $Cl^-$ to $S_{cen}$ and $S_{ext}$ does not correspond to a local free energy minimum, i.e. to a (meta-)stable state. The doubly occupied state, identified by an asterisk on *Figure 2A*, is unfavorable by 4 to 6 kCal $mol^{-1}$ in comparison to the binding of a single ion to $S_{ext}$ or to the doubly occupied configuration with ions in $S_{int}$ and $S_{ext}$.

Since current models postulate a state with simultaneous occupancy of $S_{cen}$ and $S_{ext}$, we set out to investigate what gives rise to this energetic barrier. Analyzing the conformational sampling underlying the PMF calculations, we noted that simultaneous occupancy of $S_{cen}$ and $S_{ext}$ correlated with fluctuations of the rotameric state of F357 (*Figure 2—figure supplement 1A,B*). This residue forms part of the $Cl^-$ permeation pathway of the CLC channels and transporters by coordinating ions in $S_{cen}$ and $S_{ext}$ with its backbone amide (*Park et al., 2017*). In our simulations, we find that the F357 side chain exists in equilibrium between two rotameric states with $\chi 1$ angles of $-160°$ ('up'), as seen in the crystal structures of WT and mutant CLC-ec1 (*Dutzler et al., 2002*; *Dutzler et al., 2003*), and of $-70°$ ('down') (*Figure 2E*). To test whether this conformational rearrangement affects $Cl^-$ permeation, we restrained $\chi 1$ (F357) to the 'up' or 'down' rotamers and determined the energetic landscape of ion binding in these conformations (*Figure 2C,D*). We find that when F357 is in the 'down'

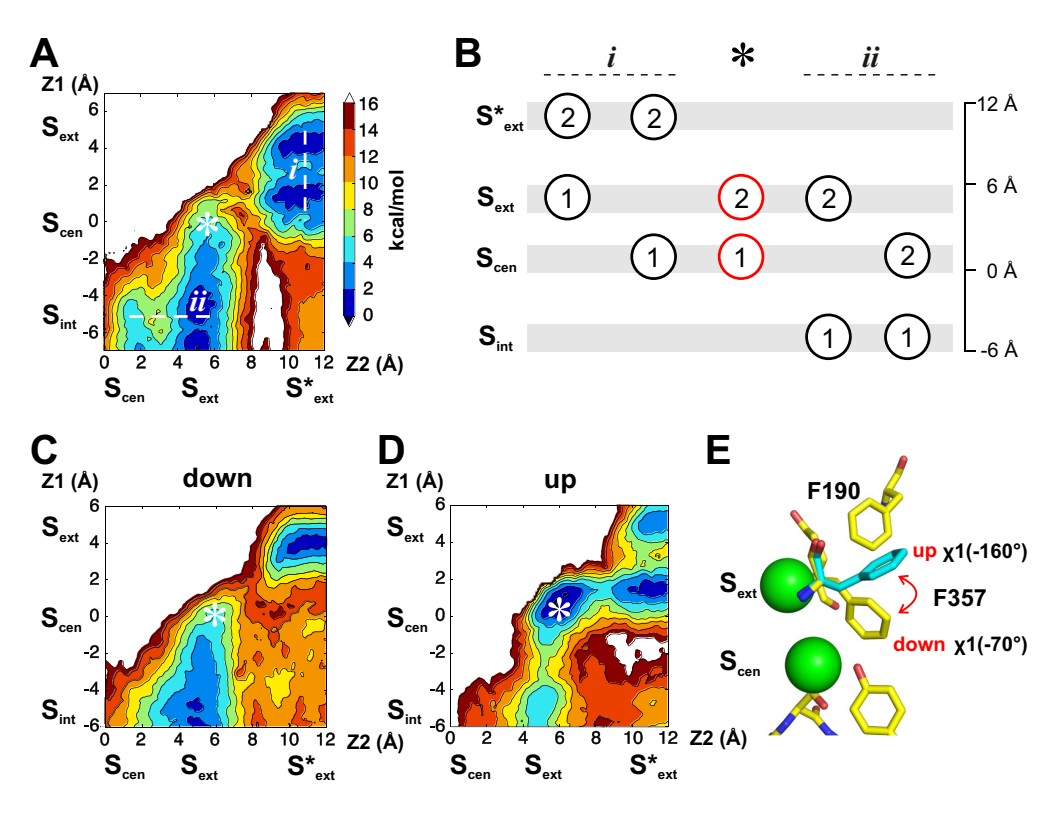

**Figure 2.** Energetic landscapes of ion movement through the permeation pathway of CLC-ec1 reveal two rotameric states of F357 that correlate with ion occupancy. (**A**) The PMF calculation describes the energetics of two Cl⁻ along the permeation pathway, with the protonated $Glu^0_{ex}$ positioned on the extracellular side of the pore (E148Q-like conformation). For both ions, the reaction coordinate consists in the position along the Z-axis (Z1, Z2). The position of the different binding sites along the Z-axis is indicated. The Z = 0 coordinate is an arbitrary point defined as the center of mass of backbone atoms around $S_{cen}$. The asterisk indicates the position of the doubly occupied state $S_{cen}/S_{ext}$. Each color of the iso-contoured map represents a dG of 2 kCal mol⁻¹, as indicated by the color scale. (**B**) The scheme illustrates the key ion occupancy states. Circles represent Cl⁻ ions identified with numbers (1, 2) that correspond to the axis labels (Z1, Z2) of the PMF plots; the labels at the top (i, ii) refer to the transitions indicated in panel A. The doubly occupied state $S_{cen}/S_{ext}$ (*) is essential for ion transport but is not observed as a stable state in the PMF calculation of panel A. (**C–D**) The PMF calculation was repeated with a harmonic restraint applied to the side chain of residue F357 to maintain χ1 around −70° ('down' conformer), (**C**) or −160° ('up' conformer), (**D**). Simultaneous Cl⁻ binding to $S_{ext}$ and $S_{cen}$ becomes energetically favorable under the 'up' conformer. Color scale is equivalent to (**A**). (**E**) F357 side chain exists in an equilibrium between two rotameric states ('up', χ1=-160°; 'down', χ1=-70°).

The online version of this article includes the following figure supplement(s) for figure 2:

**Figure supplement 1.** Ion binding influences the conformation of the F357 side chain.

**Figure supplement 2.** A crosslink known to reduce transport activity impedes the reorientation of the F357 side chain.

---

state, a free energy barrier of ~12 kCal mol⁻¹ opposes ion movement within the pore, and the $S_{cen}/S_{ext}$ double occupancy state remains unstable (**Figure 2C**). In contrast, when F357 is constrained to the 'up' conformer, two Cl⁻ ions can simultaneously bind to $S_{cen}$ and $S_{ext}$, and the different stable states along the permeation pathways are separated by free energy barriers of ~2–4 kCal mol⁻¹ (**Figure 2D**). Reciprocally, the ion occupancy state directly impacts the conformation of F357: the 'down' state is favored when no ions occupy the pathway or when only $S_{ext}$ is occupied (**Figure 2— figure supplement 1C,D**), ion occupancy of $S_{cen}$ or of $S_{cen}$ and $S_{ext}$ simultaneously favors the 'up' state of F357 (**Figure 2—figure supplement 1E,F**). This suggests that the F357 transition between the 'up' and 'down' rotamers is critical for allowing ion permeation. To test this hypothesis, we asked

whether the dynamics of F357 are affected by the introduction of a crosslink between A399 and A432 (*Figure 2—figure supplement 2A*), which inhibits transport of CLC-ec1 by preventing a conformational rearrangement involved in coupling between the intra- and extra-cellular gates (*Basilio et al., 2014*). We found that constraining the relative movements of A399 and A432 inhibits the transition of F357 between its rotamers by adding a free energy barrier of ~6 kCal mol$^{-1}$ (*Figure 2—figure supplement 2*), suggesting that this transition is part of the transport cycle. Taken together, these results suggest that the rearrangement of F357 is a critical determinant of the energy barrier height for ion movement within the CLC pore: the 'up' conformer of F357 is compatible with ion transport while its 'down' conformer disfavors ion transitions.

## Rotation of F357 enables the formation of an aromatic slide

We next asked how Cl$^-$ and Glu$_{ex}$ interact along the transport cycle. As in the previous section, we first considered states in which Glu$_{ex}$ is protonated (Glu$^0_{ex}$) and outside the Cl$^-$ pathway in an E148Q-like conformation (*Figure 3A*). Analysis of the conformational sampling of the PMFs presented in *Figure 2* reveals that, in this configuration, Glu$^0_{ex}$ is stabilized by the carboxylate group of D54 via a water molecule (*Figure 3A*) or by a hydrogen bond with the backbone of A189 (*Figure 3B*). The simulations also reveal that the carboxylate group of Glu$^0_{ex}$ rarely visits S$_{ext}$ and rather interacts with the aromatic ring of F190, even if S$_{ext}$ is free of Cl$^-$ (*Figure 3C*).

We then considered states in which Glu$_{ex}$ occupies S$_{cen}$. We calculated the PMF describing the binding of Cl$^-$ to S$_{ext}$ for both the charged and protonated forms of Glu$_{ex}$ (*Figure 3—figure supplement 1*). The PMFs show that the binding of Cl$^-$ to S$_{ext}$ requires the protonation of Glu$_{ex}$, in agreement with our previous work suggesting synergistic binding of Cl$^-$ and H$^+$ (*Picollo et al., 2012*). Interestingly, in the case of the protonated Glu$_{ex}$, a free energy well is also observed at the level of S$_{cen}$, initially occupied by the carboxylate group of Glu$^0_{ex}$. Inspection of the sampled structures reveals that, when S$_{ext}$ is occupied by a Cl$^-$, the Glu$^0_{ex}$ only partially occupies S$_{cen}$. Its carboxylate group moves sideway toward F357 and away from the Cl$^-$ pathway. Two key conformations are observed. A first one in which a proton wire composed of two water molecules forms spontaneously between Glu$^0_{ex}$ (E148) and Glu$_{in}$ (E203) (*Figure 3D*), potentially allowing the deprotonation of Glu$^0_{ex}$. In this conformation, the carboxylate group of Glu$^0_{ex}$ also forms a hydrogen bond with Y445. A second conformation reveals the possibility for Glu$^0_{ex}$ to form a π-dipole interaction with the aromatic ring of F357 (*Figure 3E*). The displacement of Glu$^0_{ex}$ toward F357 allows the bound Cl$^-$ to reach S$_{cen}$ (*Figure 3F*). The upward movement of F357 would bring the carboxylate group of Glu$^0_{ex}$ in the vicinity of F190. These calculations suggest that the conformational rearrangement of F357 enables the formation of an aromatic slide through which a protonated Glu$^0_{ex}$ can move to and from S$_{cen}$ without having to compete with the bound ions in the ion pathway.

It is important to note that in these calculations do not capture the full complexity of the energetic landscape of the interactions between ions in the CLC-ec1 pathway and F357, a task that would at minimum require a multidimensional PMF calculation involving at least three reaction coordinates: the position of the two Cl$^-$ ions and χ1(F357) angle. Additional reaction coordinates would likely be required such as the conformation and protonation state of E148, which interacts with both the ions and aromatic ring of F357, or that of conformational rearrangements taking place in CLC-ec1 during transport (*Chavan et al., 2019*). Thus, our calculations focus on the two key degrees of freedom of the system: (1) how the conformation of F357 affects ion occupancy of the CLC-ec1 pathway (*Figure 2*), and (2) how ion occupancy influences the rotameric arrangement of F357 (*Figure 2—figure supplement 1*). Importantly, while both conformers of F357 are accessible on the time scale of the simulation, χ1(F357) is a slow degree of freedom (*Figure 2—figure supplement 1A,B*), so that that the transition along this degree of freedom is not fast enough to be thoroughly sampled in absence of a biasing potential, especially when ions enter or exit the S$_{cen}$ and S$_{ext}$ binding sites. Thus, special care must be taken in interpreting the sampling of the conformations of F357 (*Figure 2—figure supplement 1A,B*).

While our finding that an inhibitory crosslink (*Basilio et al., 2014*) prevents the rearrangement of F357 (*Figure 2—figure supplement 2*), provides an initial validation of the mechanistic inferences of our MD simulations, the hypothesis that F190 and F357 play a critical role in determining Cl$^-$/H$^+$ coupling and control a rate-limiting barrier for ion transport in CLC-ec1 requires experimental probing.

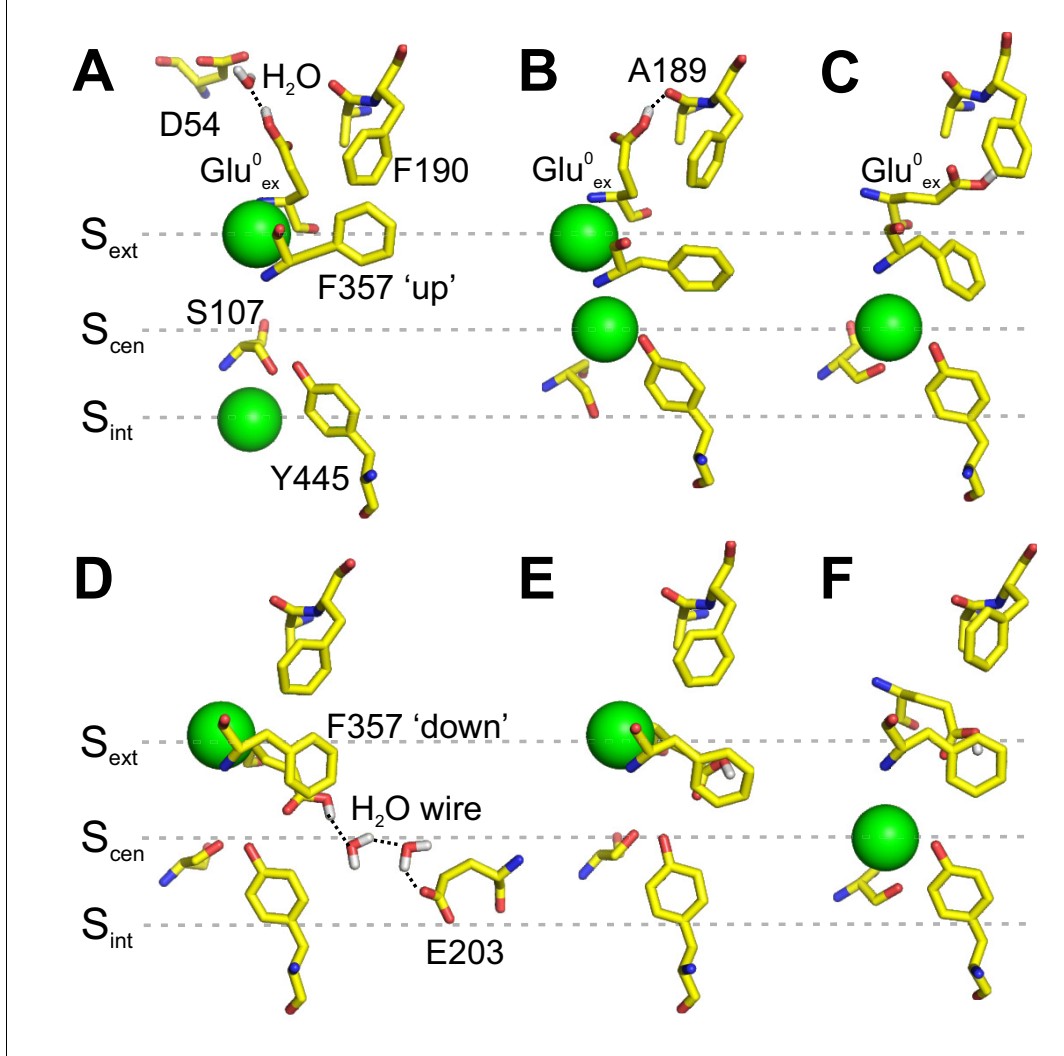

**Figure 3.** Interaction of $Glu_{ex}$ with F190 and F357. (A–C) Conformations of the pore extracted from the PMF calculation presented in *Figure 2A*, in which $Glu^0_{ex}$ is initially placed on the extracellular side of the pore. $Glu^0_{ex}$ is part of a hydrogen bond network involving D54 (A) or A189 (B). $Glu^0_{ex}$ can also interact with the side chain of F190 (C). (D–F) Conformations of the pore extracted from a 1D PMF describing the binding of a $Cl^-$ to the pore when $Glu^0_{ex}$ is initially bound to $S_{cen}$ (see *Figure 3—figure supplement 1*). A proton wire is spontaneously formed between $Glu_{ex}$ (E148) and $Glu_{in}$ (E203), which are bridged by two water molecules (D). $Glu^0_{ex}$ can form a dipole-$\pi$ interaction with F357 aromatic side chain, leaving $S_{cen}$ empty (E). $Cl^-$ moves from $S_{ext}$ to $S_{cen}$, while the side chain of $Glu^0_{ex}$ is stabilized outside the ion pathway by its interaction with F357, and in proximity of F190 (F). The snapshots shown here appear spontaneously in PMF calculations in which the position of ions was restrained. The illustrated side chain and water wire conformations are observed in at least three sampling windows and remain stable for most of the 500 ps sampling time of a given window.

The online version of this article includes the following figure supplement(s) for figure 3:

**Figure supplement 1.** Binding of a $Cl^-$ to $S_{ext}$ and $S_{cen}$ with $Glu_{ex}$ in the pore.

## The aromatic slide residues are essential for $Cl^-$:$H^+$ coupling and exchange in CLC-ec1

To test these hypotheses, we mutated them to alanine and determined the unitary transport rate and stoichiometry of the $Cl^-/H^+$ exchange cycle. Both mutations slow the turnover rate and degrade the exchange stoichiometry (*Figure 4*). The F190A mutant slows transport ~4 fold (*Figure 4B,E,F*; *Figure 4—figure supplement 1C*), while it severely impairs the transport stoichiometry to ~8.6:1 (*Figure 4D*, *Figure 4—figure supplement 1C*). The F357A mutant reduces the transport rate ~9

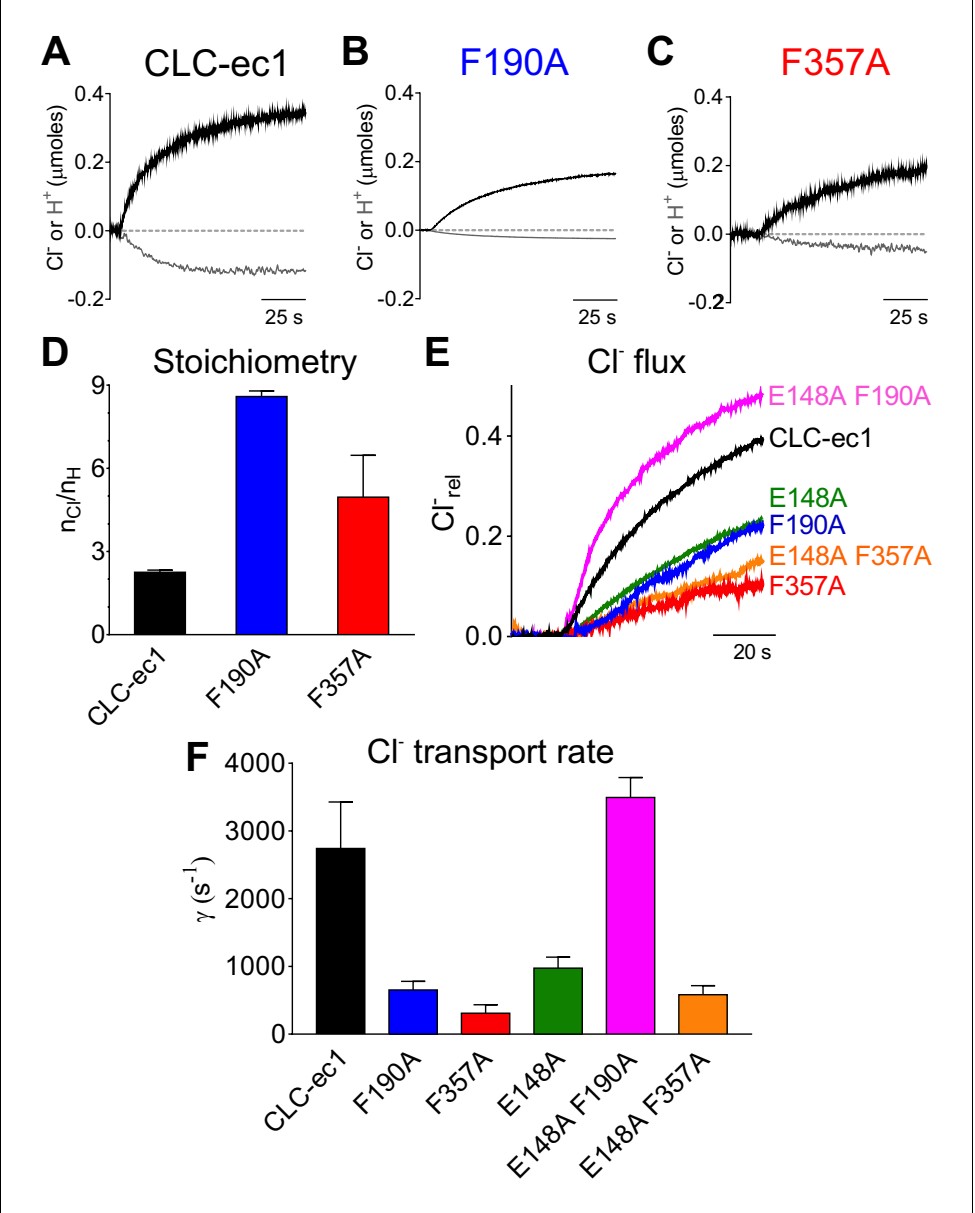

**Figure 4.** $Phe_{cen}$ (F357) and $Phe_{ex}$ (F190) determine the transport rate and $Cl^-/H^+$ coupling stoichiometry in CLC-ec1. (**A–C**) Representative time course of $Cl^-$ (black) and $H^+$ (gray) transport recordings of purified CLC-ec1 WT (A), F190A (B) and F357A (C) reconstituted into liposomes. (**D**) Average transport stoichiometry of CLC-ec1 WT (black), F190A (blue) and F357A (red). (**E**) Representative time course of $Cl^-$ efflux from proteoliposomes reconstituted with CLC-ec1 WT (black), E148A (green), F190A (blue), F357A (red), E148A F190A (pink) and E148A F357A (orange). (**F**) Average $Cl^-$ transport rate of WT and mutant CLC-ec1. All values are shown as mean ± S.E.M. and reported together with the number of repeats (N) in *Figure 4—figure supplement 1C*. The raw data for the traces shown is available in *Figure 4—source data 1*.

The online version of this article includes the following source data and figure supplement(s) for figure 4:

**Source data 1.** Time course of the representative traces.
**Figure supplement 1.** Effect of mutations on ion binding and transport in the CLC-ec1 exchanger.
**Figure supplement 2.** Conservation of $Phe_{ex}$ and $Phe_{cen}$ within the CLC family.

fold and alters the $Cl^-{:}H^+$ stoichiometry to ~4.3:1 (*Figure 4C–F*, *Figure 4—figure supplement 1C*). As the F357 backbone lines $S_{cen}$ and $S_{ext}$ (*Dutzler et al., 2002*), we tested whether an alanine substitution affects the integrity of the binding sites in two ways. First, we used isothermal titration

calorimetry (ITC) to measure Cl⁻ binding to the F357A mutant and found that it has a WT-like affinity of ~0.7 mM (*Figure 4—figure supplement 1A*). Second, we introduced the F357L substitution, which preserves side chain hydrophobicity and volume but removes the aromatic ring. We found that F357L affects the turnover rate and exchange stoichiometry like F357A (*Figure 4—figure supplement 1B,C*). Thus, the effects of the F357A mutant reflect the absence of the aromatic side chain rather than a structural disruption of the ion pathway. Taken together our functional results are consistent with the insights from our MD simulations and indicate that F190 and F357 play a key role in coupling and transport of CLC-ec1 by modulating the movement of Glu$_{ex}$ in and out of the pathway. To test this conclusion, we reasoned that the F190A and F357A mutations should not impair transport when introduced on the background of the H⁺-uncoupled E148A mutant (*Accardi and Miller, 2004*). The E148A mutant alone reduces Cl⁻ turnover ~3 fold compared to WT to ~974 ion s$^{-1}$ (*Walden et al., 2007*; *Figure 4F*). The E148A/F357A mutant transports Cl⁻ at a similar rate of the parent E148A construct, ~582 ion s$^{-1}$ (*Figure 4F*). Remarkably, the double E148A/F190A mutant 'rescues' the transport defect of the E148A parent single mutants and has a WT-like transport rate of 3,494 ions s$^{-1}$ (*Figure 4F*). The non-additivity of the effects of the F357A and F190A mutations with that of the E148A mutant qualitatively supports the idea that these Phe residues may interact with Glu$_{ex}$ during transport. Notably, F190 and F357 are among the highest conserved residues throughout the CLC family, respectively at ~94% and ~76% (*Figure 4—figure supplement 2*), suggesting that their functional role might be evolutionarily conserved. In the remainder of this work we will refer to these residues across different CLC homologues as Phe$_{ex}$ (F190 in CLC-ec1) and Phe$_{cen}$ (F357 in CLC-ec1).

## The role of Phe$_{ex}$ and Phe$_{cen}$ is conserved in mammalian transporters

We asked whether the role of Phe$_{ex}$ and Phe$_{cen}$ is conserved in the mammalian CLC exchangers CLC-7 and CLC-5. Alanine substitutions of CLC-7 Phe$_{ex}$ (F301A) and Phe$_{cen}$ (F514A) reduced the amplitude of outward currents; the current at +90 mV is reduced by ~25% for F301A and by ~50% for F514A compared to WT (*Figure 5D*). These effects could reflect either a reduction in turnover, as caused in CLC-ec1 by the corresponding mutations, or a reduced plasma membrane expression of the mutant transporters. Both mutations also left-shifted the voltage dependence of CLC-7 (*Figure 5E*); with the F301A mutant decreasing rectification so that measurable currents could be seen at negative potentials (*Figure 5*, *Figure 5—figure supplement 1*). The F301A mutant currents respond to changes in extracellular Cl⁻ and H⁺ concentrations (*Figure 5—figure supplement 1B*). A 6-fold reduction of [Cl⁻]$_{ex}$ from ~103 mM to ~17 mM induced an ~20 mV right-shift in the reversal potential (*Figure 5—figure supplement 1B*), consistent with the idea that these are Cl⁻ currents. Conversely, a 10-fold decrease in [H⁺]$_{ex}$ (from pH 7.5 to pH 8.5) only mildly affected the reversal potential (< −10 mV; *Figure 5—figure supplement 1B*), consistent with the degraded coupling stoichiometry seen in the corresponding F190A mutant in CLC-ec1 (*Figure 4D*). The inherent difficulties in controlling intracellular Cl⁻ and H⁺ concentration in oocytes prevented us from determining the exchange stoichiometry for WT and mutant CLC-7. While CLC-7 WT currents do not reach steady-state even after a 2 s activation pulse (*Figure 5A*), activation of the F301A and F514A mutants is accelerated, enabling a more thorough characterization of their voltage dependence (*Figure 5B,C*; *Figure 5—figure supplement 1C,D*). The slow activation kinetics of WT CLC-7 prevent a precise determination of its V$_{0.5}$, an osteopetrosis-causing mutant with accelerated kinetics was used to estimate the V$_{0.5}$ of CLC-7 to around +80 mV (*Leisle et al., 2011*). The F301A mutant shows a reduced voltage dependence, with an apparent $P_{minx00A0}$ of ~0.8 (*Figure 5F*, Suppl. File 1). In contrast, the F514A shows a sigmoidal voltage dependent G-V which could be well fit with a Boltzmann function, with z ~1 and V$_{0.5}$ ~+41 mV (*Figure 5F*, Suppl. File 1), consistent with the left-shift seen in the normalized I-V curves (*Figure 5E*). The corresponding mutations in the ClC-5 exchanger, F255A (Phe$_{ex}$) and F455A (Phe$_{cen}$), have qualitatively similar effects to those seen in CLC-7: the F255A mutant reduces voltage-dependent gating, resulting in measurable currents at negative voltages, and the activation threshold of the F455A mutant is left-shifted (*Figure 5—figure supplement 2*). However, the fast gating kinetics and strong rectification of this homologue prevent a reliable determination of the G-V relationship and thus a quantification of the effects. Taken together, our results are consistent with the idea that Phe$_{ex}$ and Phe$_{cen}$ play a key role in regulating the movement of Cl⁻ ions and Glu$_{ex}$ in and out of the anion pathway. This hypothesis is also consistent with the finding that the voltage-dependence of mammalian CLC exchangers arises from the movement of permeating

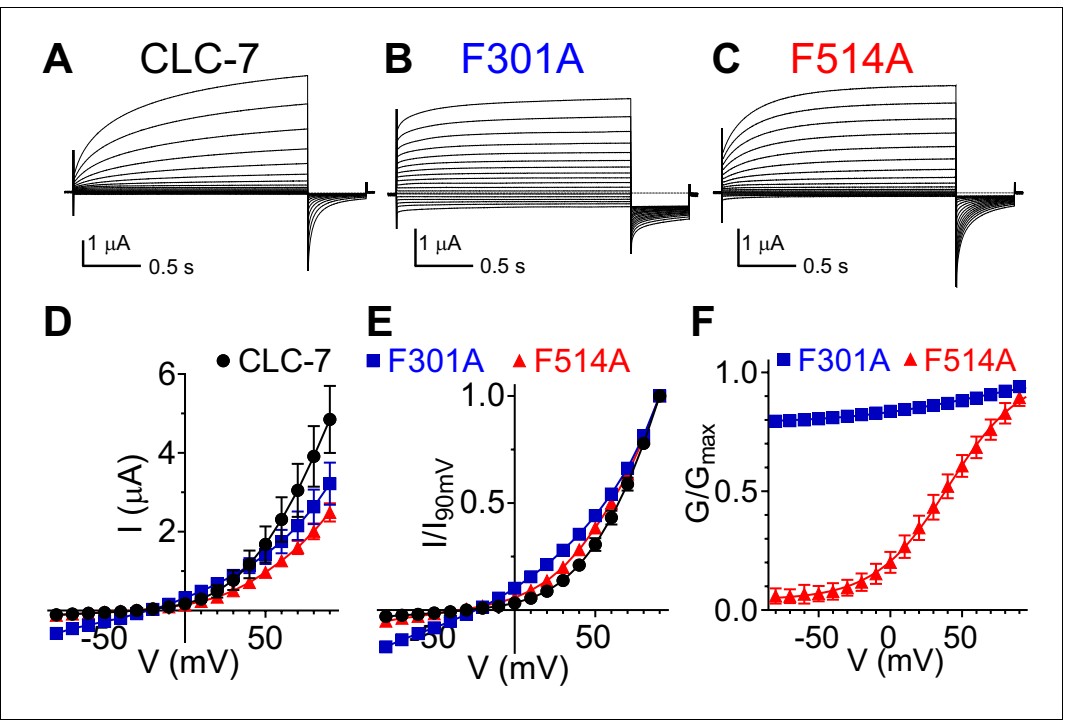

**Figure 5.** Role of Phe_cen (F514) and Phe_ex (F301) in the CLC-7 exchanger. (A–C) Representative TEVC current recordings of CLC-7 WT (A), F301A (B) and F514A (C). For voltage clamp protocol see *Methods* section. (D) I-V relationships extracted from currents at the end of the test voltage for CLC-7 WT (black), F301A (blue) and F514A (red). Symbols represent the average of independent experiments (N(WT)=10; N(F301A)=18; N(F514A)=12 oocytes from 4 to 5 batches). Solid line holds no theoretical meaning. The mean current amplitudes at +90 mV are I(WT) =4.9 ± 0.9 μA; I(F301A)=3.2 ± 0.5 μA and I(F514)=2.5 ± 0.2 μA. (E) I-V relationships from (D) normalized to corresponding I at +90 mV present changes in voltage dependence of F301A and F514A compared to WT. (F) G-V relationships for mutant CLC-7 determined from the initial values of tail currents (see *Methods*). Symbols represent the average of 12–18 independent experiments (as in (D)). Solid line is a fit to a Boltzmann function with an offset. Values are reported as mean ± S.E.M, error bars are not shown where they are smaller than the symbol size. Values for the fit parameters and number of repeats are reported in *Supplementary file 1*. The raw data for the traces shown is available in *Figure 5—source data 1*.

The online version of this article includes the following source data and figure supplement(s) for figure 5:

**Source data 1.** Individual raw data points.
**Figure supplement 1.** Impact of Phe_cen (F514) and Phe_ex (F301) mutagenesis on CLC-7 exchanger function.
**Figure supplement 1—source data 1.** Individual raw data points.
**Figure supplement 2.** Phe_cen (F455) and Phe_ex (F255) regulate the voltage dependence of the CLC-5 exchanger.
**Figure supplement 2—source data 1.** Individual normalized data points.

ions and of Glu_ex through the electric field of the transport pathway (*Smith and Lippiat, 2010*; *Zifarelli et al., 2012*).

## Phe_ex and Phe_cen are important for CLC channel gating

To probe whether the aromatic slide also plays a similar role in determining the voltage dependent gating of the CLC channels, we mutated Phe_ex (F214) or Phe_cen (F418) to alanine in the prototypical CLC-0 channel. Opening of CLC-0 channels is regulated by two processes: individual pores open and close independently in a process called single-pore gating and cooperatively during common-pore gating (*Accardi, 2015*; *Jentsch and Pusch, 2018*). Single-pore gating is thought to entail rearrangements of Glu_ex similar to those underlying the exchange cycle of the transporters, while the mechanistic underpinnings of common-pore gating remain poorly understood (*Accardi, 2015*; *Jentsch and Pusch, 2018*). In the CLC-0 channel, single-pore gating is activated by depolarizing voltages (*Figure 6A,D*), while common-pore gating is hyperpolarization activated and occurs on much

slower time scales (*Figure 6E,H*). In the F214A and F418A mutants, the single-pore gate exhibits nearly voltage-independent open probabilities over the voltage range tested (*Figure 6B–D*), suggesting that these mutations favor the open conformation of $Glu_{ex}$ (E166 in CLC-0). The voltage dependence of the common-pore gate is also affected in these mutants, as the minimal open probability increases, the gating charge decreases and $V_{0.5}$ shifts to more positive values (*Figure 6E–H*). These results suggest that $Phe_{ex}$ and $Phe_{cen}$ are shared determinants of the voltage dependence of the common- and single-pore gating processes of CLC-0. These findings are consistent with the results of the MD simulations on the CLC-ec1 exchanger that the carboxyl group of $Glu^0_{ex}$ interacts with the aromatic-slide residues to enter the pathway (*Figure 3*). Lastly, we note that the qualitative effects of the $Phe_{ex}$ mutants are conserved between CLC-7, CLC-5 and CLC-0, as in all three homologues the voltage dependence of the currents at negative voltages is drastically reduced (*Figure 5B,E,F*; *Figure 5—figure supplement 2B,D*; *Figure 6B,D*). However, we cannot exclude that these observed effects are due to indirect effects of the alanine substitutions on the CLC proteins. Indeed, the interpretation of the effects of any mutation is often complicated by the multiple simultaneous changes in side-chain properties introduced by the substitution.

## Atomic mutagenesis of the aromatic slide in CLC-0

To circumvent the limitations of alanine substitutions, we used non-canonical amino acid (ncAA) mutagenesis to selectively manipulate specific properties of the phenylalanines comprising the aromatic slide (*Figure 7*) using three derivatives: Cyclohexylalanine (Cha), 2,6diFluoro-Phenylalanine (2,6F$_2$-Phe) and 2,6diMethyl-Phenylalanine (2,6diMeth-Phe; *Figure 7A*). In Cha, the benzene ring is replaced by the non-aromatic cyclohexane ring, rendering the side chain electroneutral, while minimally altering size and hydrophobicity (*Mecozzi et al., 1996*; *Ahern et al., 2006*; *Figure 7A*). In 2,6F$_2$-Phe, the fluorine atoms at positions C2 and C6 withdraw π electrons from the face of the

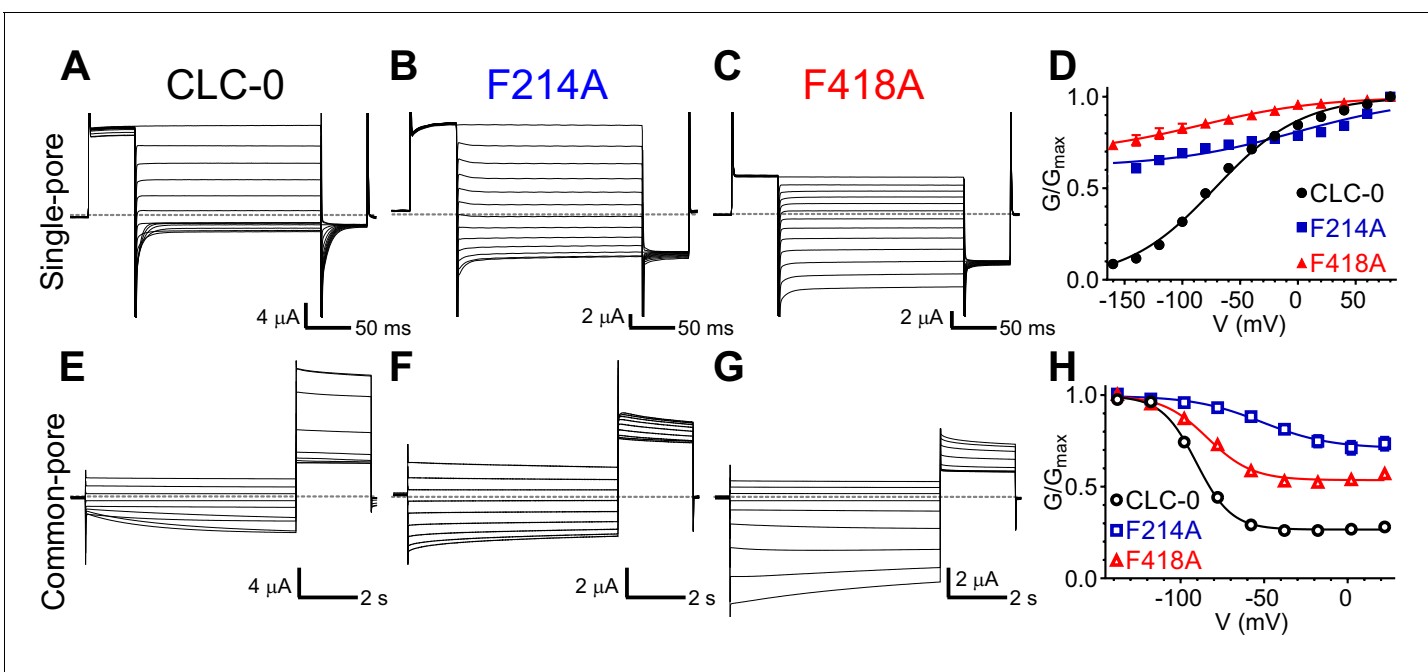

**Figure 6.** $Phe_{cen}$ (F418) and $Phe_{ex}$ (F214) contribute to voltage dependence of single- and common-pore gate of the CLC-0 channel. (A–C, E–G) Representative Two Electrode Voltage Clamp (TEVC) current recordings of CLC-0 WT (**A, E**), F214A (**B, F**) and F418A (**C, G**) evoked by single-pore (**A-C**) or common-pore (**E-G**) gating protocols (see *Methods*). (**D, H**) Normalized G-V relationships of the single- (D, filled symbols) and common-pore (H, empty symbols) gating processes of CLC-0 WT (black circles), F214A (blue squares) and F418A (red triangles). Solid lines represent fits to a Boltzmann function (*Equation 1*). Values are reported as mean ± S.E.M, error bars are not shown where they are smaller than the symbol size. Values for fit parameters and number of repeats for all conditions are reported in *Supplementary file 1*. The raw data for the traces shown is available in *Figure 6— source data 1*.

The online version of this article includes the following source data for figure 6:

**Source data 1.** Individual data points.

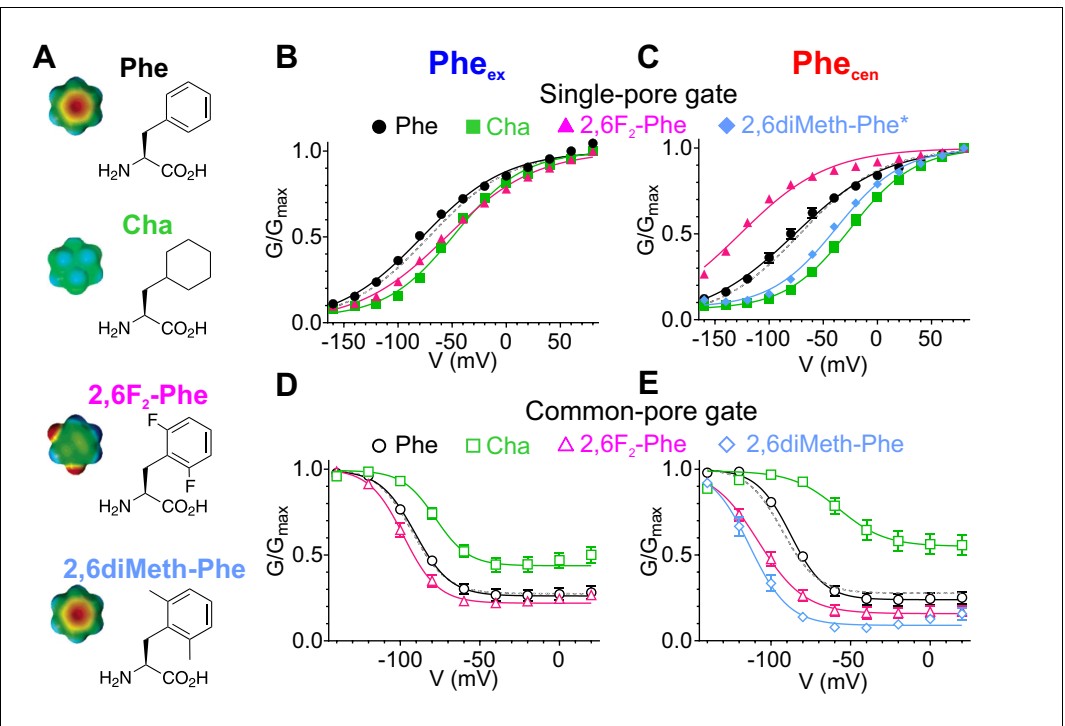

**Figure 7.** Atomic scale mutagenesis confirms importance of π-electron distribution and rotational movement of $Phe_{cen}$ for CLC-0 gating processes. (**A**) Phenylalanine (Phe) and non-canonical Phe derivatives used in this study: *Cha*, Cyclohexylalanine; *2,6F$_2$-Phe*, 2,6Fluoro-Phenylalanine; *2,6diMeth-Phe*, 2,6diMethyl-Phenylalanine. Right panel: stick representation of the amino acids, left panel: surface electrostatic potential of benzene and its derivatives with red and blue corresponding to −20 and +20 kCal mol$^{-1}$, respectively (*Mecozzi et al., 1996*). The surface electrostatic potential of 2,6diMeth-Phe is assumed similar to Phe because methyl group substitutions do not withdraw electrons from the benzene ring. (**B, C**) Normalized G-V relationships of the single- (**B, C**) and common-pore (**D, E**) gating processes for CLC-0 with the following replacements at $Phe_{ex}$ (**B, D**) and $Phe_{cen}$ (**C, E**): Phe (black circles), Cha (green squares), 2,6F$_2$-Phe (pink triangles) and 2,6diMeth-Phe (cyan diamonds). WT G-V curves (from *Figure 6*) are shown as gray dashed lines for reference. Solid lines represent fits to a Boltzmann function with an offset (see Materials and methods, *Equation 1*). Note that the G-V data for F418X+2,6diMeth-Phe was obtained on the background of the C212S mutant (2,6diMeth-Phe*) to isolate its effects on the single-pore gating process. The effects of F418X+2,6diMeth-Phe substitution on the WT background are shown in *Figure 7— figure supplement 2*. Values are reported as mean ± S.E.M, error bars are not shown where they are smaller than the symbol size. Values for the fit parameters and number of repeats are reported in *Supplementary file 1*. The raw data is available in *Figure 7—source data 1*.

The online version of this article includes the following source data and figure supplement(s) for figure 7:

**Source data 1.** Individual data points.
**Figure supplement 1.** Effects of charge redistribution at $Phe_{ex}$ and $Phe_{cen}$ on the single- and common-pore gating processes of CLC-0.
**Figure supplement 2.** Effects of CLC-0 F418-2,6diMeth-Phe on single- and common-pore gating processes.
**Figure supplement 2—source data 1.** Individual data points.
**Figure supplement 3.** Site-specific incorporation of non-canonical amino acids into CLC-0 channels is efficient and yields robust currents.
**Figure supplement 3—source data 1.** Individual data points.

---

benzene ring which makes the edges close to the backbone electronegative, the distal edges electropositive and neutralizes the negative face of the aromatic ring (*Mecozzi et al., 1996*; *Ahern et al., 2006*; *Figure 7A*). Importantly, the hydrophobicity of the residue is not affected as benzene and hexa-fluoro-benzene have similar logP values (*Leo et al., 1971*). Substitutions with these two non-canonical amino acids specifically alter the π-electron distribution of $Phe_{cen}$ and $Phe_{ex}$, thus testing the role of their electrostatic interactions with $Glu_{ex}$. The methyl groups at positions C2 and C6 in 2,6diMeth-Phe (*Figure 7A*) restrict the rotameric conversion of the aromatic side chain

around the $\chi 1$ angle (*Harrison et al., 2003*; *Li et al., 2007*), allowing us to explicitly test the role of the rotation of $Phe_{cen}$ in CLC gating.

Neutralization of electrostatics at $Phe_{cen}$ with Cha right-shifts the single-pore G-V by ~50 mV, while the redistribution of the electrons in $2,6F_2$-Phe left-shifts it by ~50 mV (*Figure 7C*, *Supplementary file 1*). These replacements have smaller effects at $Phe_{ex}$ as both the F214Cha and the F214-$2,6F_2$-Phe substitutions cause an ~20–25 mV right-shift in the G-V (*Figure 7B*, *Supplementary file 1*). Similarly, substitutions at $Phe_{cen}$ also affect common-pore gating (*Figure 7E*, *Supplementary file 1*) while those at $Phe_{ex}$ cause only minor alterations (*Figure 7D*, *Supplementary file 1*). Notably, the effect of F418Cha is comparable to that of F418A, suggesting that the electrostatics dominate the interactions of $Phe_{cen}$ with $Glu_{ex}$ during common-pore gating (*Figures 6H* and *7E*, *Supplementary file 1*). These substitutions suggest that the π-electron distribution of $Phe_{cen}$ plays a major role in channel gating, while the steric and hydrophobic nature of the phenylalanine side chain appears to play a bigger role at $Phe_{ex}$ (*Figure 7*, *Figure 7—figure supplement 1*).

Finally, to test whether the interconversion between the 'up' and 'down' rotamers of $Phe_{cen}$ plays a role in channel gating we used the rotationally restricted isostere 2,6diMeth-Phe (*Figure 7A*). The gating kinetics of the common-pore of F418-2,6diMeth-Phe are accelerated so that they become comparable to those of the single-pore gate (*Figure 7—figure supplement 2A,B*). To separate the effects of this non-canonical substitution on the single- and common-pore gating processes we used the C212S mutant which locks the common gate open (*Lin et al., 1999*; *Figure 7—figure supplement 2C,D*). The $V_{0.5}$ of the single-pore gate of the double mutant C212S F418-2,6diMeth-Phe is shifted by ~+40 mV, while the $V_{0.5}$ of the common-pore gate of the single mutant is shifted by ~−25 mV (*Figure 7C,E*, Suppl. File 1). Thus, restricting the conformational rearrangement of $Phe_{cen}$ between the 'up' and 'down' conformers impairs gating of the CLC-0, consistent with the finding of our MD simulations on the CLC-ec1 transporter (*Figure 3*). Taken together, our results suggest that the aromatic slide formed by $Phe_{cen}$ and $Phe_{ex}$ forms an evolutionarily conserved structural motif that enables movement of $Glu_{ex}$ in and out of the $Cl^-$ pore during the exchange cycle and gating of CLC exchangers and opening of CLC channels.

## Discussion

Despite extensive structural and functional investigations, the mechanisms underlying the exchange cycle of the CLC transporters and opening of the CLC channels remain poorly understood. These processes are evolutionarily and mechanistically related (*Miller, 2006*; *Accardi, 2015*; *Jentsch and Pusch, 2018*): in both channels and transporters $Glu_{ex}$ moves in and out of the $Cl^-$ pathway in a protonation-dependent manner. However, the molecular steps that underlie these rearrangements are unclear. The CLCs exchangers do not follow the 'ping-pong' or sequential kinetics that characterize most conventional transporters. In contrast, CLCs simultaneously bind $H^+$ and $Cl^-$ (*Picollo et al., 2012*), and substrate movement occurs along two partially congruent translocation pathways (*Accardi et al., 2005*; *Zdebik et al., 2008*). Yet, how the two substrates bypass each other in the protein remains unknown. The recent structures of the $CLC^F$ exchanger and of the CLC-1 $Cl^-$ channel suggested that $Glu_{ex}$ might interact with $Phe_{cen}$ (*Last et al., 2018*; *Park and MacKinnon, 2018*), but the functional implications of these interactions are not clear. Current transport mechanisms for CLC exchangers are not reversible, as they all postulate a step where a protonated and neutral $Glu_{ex}$ displaces $Cl^-$ ions bound to $S_{ext}$ and $S_{cen}$ sites to transfer its proton to the internal solution (*Accardi et al., 2005*; *Zdebik et al., 2008*). This transition is energetically unfavorable (*Kuang et al., 2007*), and would result in highly asymmetric transport rates. However, the CLC exchangers function with comparable efficiency in both directions (*Matulef and Maduke, 2005*; *Leisle et al., 2011*; *De Stefano et al., 2013*), arguing against the existence of an asymmetric rate-limiting step. Here, a combination of MD simulations with biochemical and electrophysiological experiments suggest that $Glu_{ex}$ takes different routes depending on its protonation state, allowing for full reversibility of the transport mechanism.

### Formation of an aromatic slide is essential to $Glu_{ex}$ movement

Our data suggest that while the deprotonated and negatively charged $Glu_{ex}$ moves through the $Cl^-$ pathway, the protonated and neutral $Glu^0_{ex}$ interacts with two highly conserved phenylalanines, $Phe_{cen}$ and $Phe_{ex}$. These residues can form an aromatic slide that enables movement of protons bound

to $Glu_{ex}$ in and out of the pathway, connecting the central binding site and the extracellular solution. In the available structures of CLC channels and exchangers (*Dutzler et al., 2002*; *Dutzler et al., 2003*; *Accardi et al., 2005*; *Accardi et al., 2006*; *Lobet and Dutzler, 2006*; *Feng et al., 2010*; *Jayaram et al., 2011*; *Basilio et al., 2014*; *Park et al., 2017*; *Last et al., 2018*; *Park and MacKinnon, 2018*; *Wang et al., 2019*), $Phe_{cen}$ is in the 'up' rotamer suggesting this conformation is likely the most stable. However, our combined simulation and functional data suggest that $Phe_{cen}$ can adopt two distinct rotamers around its χ1 angle (*Figure 2*), and that this rearrangement determines the energy barrier for $Cl^-$ movement within the pore (*Figure 2*, *Figure 2—figure supplement 2*) and enables $Glu^0_{ex}$ to reach or leave $S_{cen}$ via the aromatic slide pathway (*Figure 3*). Further, we show that the aromatic slide residues play an evolutionarily conserved key role in the CLC exchange cycle and in CLC channel gating. Our conventional mutagenesis underscores the conserved importance of $Phe_{cen}$ and $Phe_{ex}$ in transporter and channel gating: alanine substitutions at these positions impair the absolute transport rate, the voltage dependence and the exchange stoichiometry of the transporters (*Figures 4* and *5*, *Figure 5—figure supplement 2*), and nearly abolish the voltage dependence of channel gating (*Figure 6*). We used atomic scale mutagenesis to test the prediction of our MD simulations that formation of the aromatic slide entails a rotation of the side chain of $Phe_{cen}$ around its $C_\alpha$-$C_\beta$ bond. When we replace $Phe_{cen}$ with 2,6diMet-Phe, a Phe derivative that specifically constrains this rotational rearrangement, we find that opening of the single- and common-pore gates in the CLC-0 channel is impaired (*Figure 7*, *Figure 7—figure supplement 2*). Thus, the aromatic slide forms an evolutionarily conserved structural motif that might enable movement of a protonated $Glu^0_{ex}$ in and out of the pathway in both CLC channels and transporters, consistent with the finding that some CLC channels mediate some residual $H^+$ transport (*Picollo and Pusch, 2005*; *Lísal and Maduke, 2008*). The low functional expression of the CLC-5 and −7 transporters encoding the non-canonical amino acid prevented a similar test in these homologues. Moreover, for CLC-ec1, the technology for incorporation of these particular unnatural amino acids is currently not available for prokaryotic expression systems.

Our results also suggest that $Phe_{ex}$ and $Phe_{cen}$ interact differently with $Glu_{ex}$. The aromatic ring of $Phe_{ex}$ appears to play a structural role in helping position $Glu_{ex}$ where it can interact with $Phe_{cen}$ in the 'up' conformation (*Figure 3*). Indeed, removal of the aromatic ring of $Phe_{ex}$ severely affects voltage dependent gating of CLC channels and transporters (*Figures 5* and *6*), while selective manipulations of its electrostatic properties have only minor effects on channel gating (*Figure 7B,D*), possibly because this residue lines a water-accessible extracellular vestibule in the CLCs. In contrast, the electrostatic properties of the aromatic ring of $Phe_{cen}$ are essential determinants for $Glu_{ex}$ movement in the CLC-0 channel: elimination of the π-electrons of $Phe_{cen}$ favors the closed state of the single-pore gate (*Figure 7C*), while their re-localization to the proximal edge promotes opening (*Figure 7C*). These findings are consistent with the location of $Phe_{cen}$ within the core of the protein where changes in electrostatics are likely to have more impact, and with our MD simulations suggesting that $Glu_{ex}$ forms a π-dipole interaction with the aromatic ring of $Phe_{cen}$ (*Figure 3*). Indeed, the interaction between the buried aromatic $Phe_{cen}$ and $Glu_{ex}$ could account for the observed shifts in the pKa of $Glu_{ex}$ (*Robinson et al., 2017*) in CLC channels and transporters to keep $Glu_{ex}$ protonated during its transition through the protein (*Hanke and Miller, 1983*; *Niemeyer et al., 2009*; *Picollo et al., 2010*; *Picollo et al., 2012*). Our results are in harmony with the recent proposal that $Glu_{ex}$ can adopt a conformation where it directly interacts with $Phe_{cen}$ in a fluoride selective CLC antiporter (*Last et al., 2018*), in a mutant mimicking a fully-protonated CLC-ec1 (*Chavan et al., 2019*) and in the CLC-1 channel (*Park and MacKinnon, 2018*; *Figure 4—figure supplement 2C*).

## A mechanism for $Cl^-$/$H^+$ exchange

Our finding that two Phe residues form an evolutionarily conserved secondary pathway that enables movement of the protonated $Glu_{ex}$ in and out of the ion transport pathway allows us to propose a 7-state mechanism for the CLC transporters that explains the stoichiometry of 2 $Cl^-$:1 $H^+$, is fully reversible, and accounts for previous results. Our findings provide structural and mechanistic grounds to explain how $Glu_{ex}$ and $Cl^-$ ions can swap places in the CLC permeation pathway (*Jentsch and Pusch, 2018*). For simplicity of representation, we consider that an ion in the $S^*_{ext}$ site is out of the pathway and as such, in our scheme we do not show ions bound to this site. As a starting configuration, we consider a state where $Glu_{ex}$ and $Glu_{in}$ are de-protonated, $Glu_{ex}$ occupies $S_{cen}$, $Phe_{cen}$ is in the 'up' position and no $Cl^-$ ions are bound to the pathway (*Figure 8*, I). After a $Cl^-$ ion binds to $S_{int}$ (*Figure 8*, II),

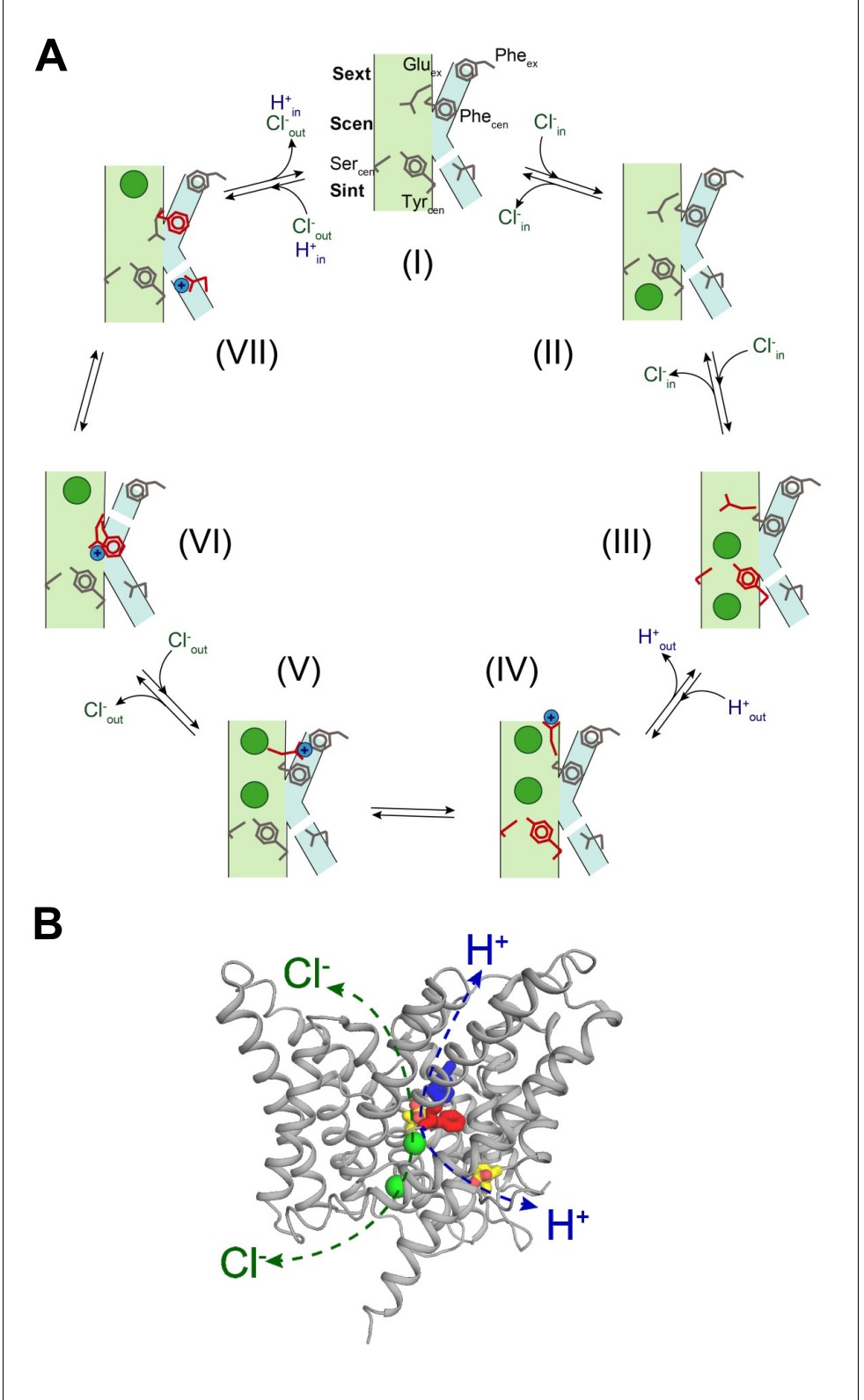

**Figure 8.** Proposed mechanism for the 2 Cl⁻: 1 H⁺ CLC exchangers. (**A**) Schematic representation of the 2 Cl⁻: 1 H⁺ CLC exchange cycle. The Cl⁻ ions are shown as green circles and H⁺ as blue circles. The Cl⁻ and H⁺ pathways are shown in pale green and cyan respectively. For clarity, the residues that undergo conformational rearrangements in each step are highlighted in red. The H⁺ pathway is shown as discontinuous when it is in a non-conductive

*Figure 8 continued on next page*

*Figure 8 continued*

conformation. (I) Apo- and occluded-state of the transporter with $Glu_{ex}$ in $S_{cen}$ and $Phe_{cen}$ in the up rotamer. (II) An intracellular $Cl^-$ ion binds to $S_{int}$. (III) The inner gate opens, a second intracellular $Cl^-$ binds so that both $S_{int}$ and $S_{cen}$ are occupied and $Glu_{ex}$ moves to $S_{ext}$. (IV) The $Cl^-$ ions move to $S_{cen}$ and $S_{ext}$, the inner gate closes, $Glu_{ex}$ moves out of the pathway and becomes protonated. (V) $Glu^0_{ex}$ interacts with $Phe_{ex}$. (VI) $Glu^0_{ex}$ interacts with $Phe_{cen}$ which rotates to the down conformation enabling the formation of a water wire connecting $Glu^0_{ex}$ and $Glu_{in}$; one $Cl^-$ ion is released from $S_{ext}$ and the other one moves from $S_{cen}$ to $S_{ext}$. (VII) The $H^+$ is transferred to $Glu_{in}$ favoring movement of $Glu_{ex}$ into $S_{cen}$ and the $Cl^-$ ion in $S_{ext}$ is released to the extracellular solution, returning to the initial state (I). (B) Physically distinct $Cl^-$ and $H^+$ pathways are indicated in CLC-ec1 WT structure ($Glu_{ex}$ and $Glu_{in}$ are shown in yellow, $Phe_{ex}$ in blue, $Phe_{cen}$ in red, $Cl^-$ ions as green spheres). Both pathways converge at $Glu_{ex}$.

opening of the intracellular gate, formed by $Ser_{cen}$ and $Tyr_{cen}$ (*Basilio et al., 2014*), allows the ion to move into $S_{cen}$, displacing $Glu_{ex}$ to $S_{ext}$ (*Figure 8*, III). Binding of a second ion to $S_{ext}$ favors protonation of $Glu_{ex}$, and is accompanied by the displacement of $Glu^0_{ex}$ out of the pathway and closure of the internal gate (*Figure 8*, IV). The protonated $Glu^0_{ex}$ diffuses toward the aromatic slide and interacts with $Phe_{ex}$ (*Figure 8*, V). Movement of $Glu^0_{ex}$ along the aromatic slide allows the interdependent rearrangement of $Phe_{cen}$ to the 'down' state and release of a $Cl^-$ ion, i.e. release from $S_{ext}$ to the extracellular milieu and the transfer of the second ion from $S_{cen}$ to $S_{ext}$ (*Figure 8*, VI). This conformation, where $Glu^0_{ex}$ interacts with $Phe_{cen}$ on the side of the $Cl^-$ pathway and $S_{ext}$ is occupied, favors the formation of a water wire (*Figure 8*, VI), which enables proton transfer from $Glu^0_{ex}$ to $Glu_{in}$ (*Figure 8*, VII). Deprotonation of $Glu_{ex}$ allows it to move into $S_{cen}$, favoring the release of the second $Cl^-$ from $S_{ext}$ to the outside, and of the proton from $Glu^0_{in}$ to the intracellular solution, returning to the starting configuration (*Figure 8*, I). We note that the $Cl^-$ binding affinity to $S_{int}$ is low as there is no free energy barrier preventing the release of the ion (*Accardi et al., 2006*; *Lobet and Dutzler, 2006*; *Picollo et al., 2009*). Thus, when the transporter is in states IV through VII it is likely that $Cl^-$ ions will bind to and unbind from $S_{int}$. However, as long as the intracellular gate remains closed these events would not participate in the permeation process. Thus, for simplicity we have omitted them from the gating scheme. In sum, we propose that the $Cl^-$ and $H^+$ ions bypass each other while moving in opposite directions through a CLC transporter by taking physically distinct routes: the $Cl^-$ ions move through the anion selective pore while the $H^+$ moves along a pathway comprised of a water-wire and the aromatic slide (*Figure 8B*). The distinct routes for $Cl^-$ and $H^+$ allow for a complete reversibility of the cycle.

Remarkably, our findings show that the role of the aromatic slide residues $Phe_{cen}$ and $Phe_{ex}$ is evolutionarily conserved between CLC exchangers and channels. We propose that the aromatic slide in CLC-0 could provide a conserved pathway that enables the residual $H^+$ transport associated with the common-pore gating process of the CLC-0 channel to occur (*Lísal and Maduke, 2008*). Indeed, the role of $Phe_{cen}$ in CLC function is also supported by the finding that mutations at this position severely affect CLC-1 channel gating and conductance (*Estévez et al., 2003*) and cause dominant myotonia (*Imbrici et al., 2015*). Thus, our proposed exchange mechanism for the CLC transporters also captures the key rearrangements that underlie gating of the CLC channels. In this framework, rapid ion conduction would be enabled by the disruption of the intracellular gate, a hypothesis justified by the widened intracellular vestibule seen in the recent structures of the bCLC-K and hCLC-1 channels (*Park et al., 2017*; *Park and MacKinnon, 2018*; *Wang et al., 2019*).

## Materials and methods

### Molecular system

The molecular systems, based on the CLC-ec1 crystal structure (PDB: 1OTS) (*Dutzler et al., 2003*), were assembled using the CHARMM-GUI web service (*Jo et al., 2008*). As it was shown that the monomeric form of CLC-ec1 is functional (*Robertson et al., 2010*), a single subunit was used. The x-ray structure includes residues 18 to 458. Residues 18 to 31 are normally interacting with the opposite subunit and were removed here to prevent any unexpected interaction with the pore. All residues are in their natural protonation state at pH seven except Glu148, which is tested in both its unprotonated and protonated forms. The orthorhombic periodic simulation cell contains a ClC-ec1 monomer inserted in a bilayer composed of 262 dimystoylphosphatidylcholine (DMPC) lipids, about

13 600 TIP3P water molecules, and potassium/chloride ions to neutralize the systems at a concentration of about 150 mM KCl.

## Potential function and simulations

All simulations were run using the CHARMM simulation software version c36 (*Brooks et al., 2009*) using the CHARMM36 force field (*Best et al., 2012*). Simulations were performed in an isothermal−isobaric ensemble with a pressure of 1 atm and a temperature of 323 K. Particle-mesh Ewald method (*Essmann et al., 1995*) was applied for the calculation of electrostatic interactions with a grid spacing of 1 Å. The cutoff distance for van der Waals interactions was taken at 12 Å with a switching function starting at 10 Å. Time step for the integration of the motion was set to two fs. The membrane systems were equilibrated following a standard protocol offered by CHARMM-GUI (*Wu et al., 2014*). After equilibration, a simulation of 100 ns was run for each molecular system prior to the free energy calculations. All MD simulations were performed at 0 mV.

Using the CHARMM36 force field, we compared the strength of a π-dipole interaction and that of a typical H-bond. We calculated that the interaction between a protonated glutamate side chain and the aromatic ring of a phenylalanine is −5.0 kCal mol$^{-1}$ (for an optimized conformation in vacuum). For comparison, the interaction between a protonated glutamate and a serine side chain is −5.8 and −7.2 kCal mol$^{-1}$, depending if the unprotonated or the protonated oxygen of the carboxylate group is involved. These number are in line with ab initio calculations that reported interactions on the order of −2.4 to −7.1 kCal mol$^{-1}$ (*Du et al., 2013*).

## Potential of mean force calculations

The potential of mean forces (PMF) were calculated using the self-learning adaptive umbrella sampling method (*Wojtas-Niziurski et al., 2013*). In the case of PMFs describing the displacement of one or two ions, the reaction coordinate is the position of a given ion in the pore, as projected on the normal to the membrane (Z axis). The reference point is the center of mass of the backbone of residues 107, 356, 357, and 445. Independent simulations of 500 ps were performed every 0.5 Å along the reaction coordinate using a biasing harmonic potential of 20 kCal/mol•Å$^2$. For the 2D PMFs of *Figure 2*, the ions were initially placed in S$_{cen}$ and S$_{ext}$. For the 1D PMF of *Figure 3—figure supplement 1*, the ion was placed in S$_{ext}$. In the case of PMFs describing the rotation of the F357 side chain, the reaction coordinate is the $\chi 1$ angle. Simulations were performed every 10 degrees using a biasing harmonic potential of 20 kCal/mol•rad$^2$. In all cases the first 50 ps of every independent window simulation was considered as equilibration and thus excluded from the analysis. The independent simulations were combined and unbiased using the weighted histogram analysis method (WHAM) (*Kumar et al., 1992*). The conformations shown in *Figure 3* are observed in at least three sampling windows and remain stable for most of the sampling time of a given window.

## Protein purification and liposome reconstitution

Expression and purification of wild-type and mutant CLC-ec1 were performed according to published protocols (*Accardi and Miller, 2004*; *Accardi et al., 2006*; *Picollo et al., 2009*; *Picollo et al., 2012*; *Basilio et al., 2014*). Purified proteins were reconstituted into liposomes as described (*Basilio et al., 2014*).

## Cl$^-$ and H$^+$ flux recordings for CLC-ec1 and variants

Cl$^-$ and H$^+$ fluxes of CLC-ec1 wild-type and mutant proteins reconstituted into proteoliposomes were recorded simultaneously and the coupling stoichiometry was determined as described (*Basilio et al., 2014*).

## Isothermal titration calorimetry (ITC)

Cl$^-$ binding affinity was determined for purified CLC-ec1 F357A as described (*Picollo et al., 2009*; *Basilio et al., 2014*) using a nanoITC instrument (TA Instruments). For these experiments, the final purification step of the protein was purified over a gel filtration column pre-equilibrated in 100 mM Na-K-Tartrate, 20 mM Hepes, 50 μM DMNG, pH 7.5 (Buffer B0) and concentrated to 50–195 μM. The injection syringe was filled with buffer B0 with 50 mM KCl added, to achieve final Molar Ratios of 70–100. Each experiment consisted of 30–48 injections of 1 μl of the ligand solution at 3–4 min

intervals into the experimental chamber kept under constant stirring at 350 rpm and at 25.0 ± 0.1°C. All solutions were filtered and degassed prior to use. The ITC data was fit to a single site Wiseman isotherm using the NanoAnalyze program from TA instruments.

## In vitro cRNA transcription

RNAs for all CLC-0, hCLC-7, hCLC-5 and mOstm1 wild-type and mutant constructs were transcribed from a pTLN vector using the mMessage mMachine SP6 Kit (Thermo Fisher Scientific, Grand Island, NY) (*Pusch et al., 1995*; *Steinmeyer et al., 1995*; *Leisle et al., 2011*). For all experiments in this paper a plasma-membrane localized version of hCLC-7 has been used as 'wild-type' which has been published earlier termed as CLC-7$^{PM}$ (*Leisle et al., 2011*). For final purification of cRNA the RNeasy Mini Kit (Quiagen, Hilden, Germany) was employed. RNA concentrations were determined by absorbance measurements at 260 nm and quality was confirmed on a 1% agarose gel.

## tRNA misacylation

For nonsense suppression of CLC-0 TAG mutants (F214X and F418X) in *Xenopus laevis* oocytes, THG73 and PylT tRNAs have been employed. THG73 was transcribed, folded and misacylated as previously described (*Leisle et al., 2016*). PylT was synthetized by Integrated DNA Technologies, Inc (Coralville, IA, USA), folded and misacylated as previously described (*Infield et al., 2018*). Phe-, Cha-, 2,6F2-Phe- and 2,6diMethPhe-pdCpA substrates were synthesized according to published procedures (*Infield et al., 2018*). L-Cha was purchased from ChemImpex (Wood Dale, IL, USA; catalogue number: 11696) and BACHEM (USA; catalogue number: F-2500.0001), L-2,6-difluoro Phe from ChemImpex (catalogue number: 24171) and L-2,6-dimethyl Phe from Enamine (Monmouth Junction, NJ, USA; catalogue number: EN300-393063).

## Protein expression in *Xenopus laevis* oocytes and two electrode voltage clamp (TEVC) recordings

*Xenopus laevis* oocytes were purchased from Ecocyte Bio Science (Austin, TX, USA) and Xenoocyte (Dexter, Michigan, USA) or kindly provided by Dr. Pablo Artigas (Texas Tech University, USA, protocol # 11024). For conventional CLC expression, following injection and expression conditions have been used: for CLC-7 (WT, F301A or F514A) and Ostm1, 25–75 ng of each cRNA were injected per oocytes and currents were recorded after ~60–80 hr; for CLC-5 (WT, F255A or F445A), 50 ng cRNA were injected per oocyte and currents were recorded ~48–72 hr; for CLC-0 (WT, F214A or F418A), 0.1–5 ng cRNA were injected and currents were measured ~6–24 hr after injection. For nonsense suppression of CLC-0 constructs (F214X, F418X, C212S F418X), cRNA and misacylated tRNA were coinjected (up to 25 ng of cRNA and up to 250 ng of tRNA per oocyte) and currents were recorded 6–24 hr after injection.

TEVC was performed as described (*Picollo et al., 2009*). In brief, voltage-clamped chloride currents were recorded in ND96 solution (in mM: 96 NaCl, 2 KCl, 1.8 CaCl2, 1 MgCl2, 5 HEPES, pH 7.5) using an OC-725C voltage clamp amplifier (Warner Instruments, Hamden, CT). The data was acquired with Patchmaster (HEKA Elektronik, Lambrecht, Germany) at 5 kHz (CLC-0) and 20 kHz (CLC-5, CLC-7) and filtered with Frequency Devices 8-pole Bessel filter at a corner frequency of 2 kHz. Analysis was performed using Ana (M. Pusch, Istituto di Biofisica, Genova) and Prism (GraphPad, San Diego, CA, USA). Standard voltage-clamp protocols have been applied for the three CLC proteins, the holding potential was constant at −30 mV. For CLC-0 two different recording protocols have been used to distinguish single-pore from common-pore gating. During the single-pore gating protocol the voltage was stepped to +80 mV for 50 ms and then a variable voltage from −160 mV to +80 mV increasing in 20 mV steps was applied for 200 ms, followed by a 50 ms pulse at −120 mV for tail current analysis. For CLC-0 common-pore gating, 7 s voltage steps from +20 mV to −140 mV have been applied in −20 mV increments followed by a 2.5 s +60 mV post pulse for tail current analysis. For CLC-5, a simple voltage step protocol was applied: 400 ms steps from −80 to + 80 mV in 20 mV increments. For CLC-7/Ostm1, voltage was clamped at variable values from −80 to +90 mV in 10 mV steps for 2 s, followed by a 0.5 s post pulse at −80 mV for tail current analysis.

To estimate the voltage dependence of CLC-0 and CLC-7/Ostm1 mutants gating, tail current analysis was performed and data was fit to a Boltzmann function of the form:

$$P_o = P_{min} + \frac{(1 - P_{min})}{1 + \exp\left[(V_{0.5} - V)/k\right]} \tag{1}$$

where $P_{o/italic}$ is the open probability as a function of voltage and is assumed to reach a value of unity at full activation. $P_{min/italic}$ is the residual open probability independent of voltage. $V_{0.5}$ is the voltage at which 50% activation occurs, and $k = RT/zF$ is the slope factor, R is the universal gas constant, T is temperature in K, F is the Faraday constant, and z is the gating charge.

For analysis of the activation kinetics of CLC-7/Ostm1 and its variants, activating voltage pulses (from +20 to +90 mV) were fit to a bi-exponential function of the following form:

$$I = A_0 + A_1 e^{-t/\tau_1} + A_2 e^{-t/\tau_2} \tag{2}$$

where I is the current as a function of time; $A_1$, $A_2$ and $A_0$ are fractional amplitudes obtained by normalizing to the total current. While $A_1$ and $A_2$ are time-dependent components, $A_0$ is time-independent. $\tau_1$ and $\tau_2$ are the corresponding time constants.

## Statistical analysis

All values are presented as mean ± s.e.m. To determine statistical significance Student's t-test (two-tailed distribution; two-sample equal variance) was performed. The threshold for significance was set to $p = 0.05$.

## Acknowledgements

The authors wish to thank members of the Accardi lab for helpful discussions. This work was supported by NIH/NIGMS grant GM128420 (to AA), Swiss National Science Foundation SNF Professorship PP00P3_139205 (to SB) and NIH/NINDS grant NS104617 (to CAA). YX was supported by the China Scholarship Council. Calculations were performed at sciCORE (http://scicore.unibas.ch) scientific computing core facility at University of Basel, and using the PRACE-3IP project (FP7 RI-312763) resource Lindgren based in Sweden at PDC.

## Additional information

### Funding

| Funder | Grant reference number | Author |
| --- | --- | --- |
| National Institutes of Health | GM128420 | Alessio Accardi |
| National Institutes of Health | NS104617 | Christopher A Ahern |
| Swiss National Science Foundation | PP00P3_139205 | Simon Bernèche |
| China Scholarship Council | | Yanyan Xu |

The funders had no role in study design, data collection and interpretation, or the decision to submit the work for publication.

### Author contributions

Lilia Leisle, Conceptualization, Data curation, Formal analysis, Validation, Investigation, Visualization, Methodology; Yanyan Xu, Data curation, Formal analysis, Investigation, Methodology; Eva Fortea, Malvin Vien, Formal analysis, Investigation; Sangyun Lee, Investigation; Jason D Galpin, Resources, Methodology; Christopher A Ahern, Conceptualization, Supervision, Funding acquisition, Investigation, Methodology; Alessio Accardi, Conceptualization, Data curation, Formal analysis, Supervision, Funding acquisition, Investigation, Project administration; Simon Bernèche, Conceptualization, Software, Supervision, Funding acquisition, Investigation, Project administration

## Author ORCIDs

Lilia Leisle (ID) https://orcid.org/0000-0003-3987-4269
Eva Fortea (ID) https://orcid.org/0000-0003-4328-8940
Christopher A Ahern (ID) http://orcid.org/0000-0002-7975-2744
Alessio Accardi (ID) https://orcid.org/0000-0002-6584-0102
Simon Bernèche (ID) https://orcid.org/0000-0002-6274-4094

## Decision letter and Author response

Decision letter https://doi.org/10.7554/eLife.51224.sa1
Author response https://doi.org/10.7554/eLife.51224.sa2

## Additional files

### Supplementary files

• Supplementary file 1. Summary of the Boltzmann fitting parameters. The following parameters from Boltzmann fits for CLC-7 and CLC-0 constructs used in *Figures 5–7* are reported as mean ± S. E.M.: $p_{min}$, minimal open probability; $V_{0.5}$, voltage of half maximal activation; $z$, gating charge; $N$, number of oocytes; $n$, number of independent oocyte batches. * indicates values of the fit parameters that were not well constrained during fitting, as such they should be considered as estimates of the parameters.

• Transparent reporting form

### Data availability

All data generated or analyzed during this study are included in the manuscript and supporting files. The data files for the MD simulations presented in Figure 2, Figure 3, and associated figure supplements were deposited in the figshare repository under the following identifier: https://doi.org/10. 6084/m9.figshare.12116784. The raw data files for the representative electrophysiological and ion flux traces have been uploaded as source data files as indicated in the figure legends. All additional data are available upon request.

The following dataset was generated:

| Author(s) | Year | Dataset title | Dataset URL | Database and Identifier |
|---|---|---|---|---|
| Leisle L, Xu Y, Fortea E, Lee S, Galpin JD, Vien M, Ahern CA, Accardi A, Bernèche S | 2020 | Molecular dynamics free energy calculations highlighting the role of F357 in the function of the CLC-ec1 Cl-/H+ antiporter | https://doi.org/10.6084/m9.figshare.12116784 | figshare, 10.6084/m9.figshare.12116784 |

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
