## [Decision Letter]

**Acceptance summary:**

Chloride-hydrogen exchangers of the CLC family are important targets for known human disorders. Unraveling their molecular mechanisms is important for understanding their role in physiology. In this manuscript the authors show a novel mechanism for the simultaneous occupancy of chloride and protons in the transport pathway. These results should open new avenues of research to understand CLC transporters and channels.

**Decision letter after peer review:**

Thank you for submitting your article "Divergent Cl^-^ and H^+^ pathways underlie transport coupling and gating in CLC exchangers and channels" for consideration by *eLife*. Your article has been reviewed by three peer reviewers, including Leon D Islas as the Reviewing Editor and Reviewer #1, and the evaluation has been overseen by a Reviewing Editor and Richard Aldrich as the Senior Editor. The following individuals involved in review of your submission have agreed to reveal their identity: Michael Pusch (Reviewer #2); Merritt Maduke (Reviewer #3).

The reviewers have discussed the reviews with one another and the Reviewing Editor has drafted this decision to help you prepare a revised submission. A fourth expert on MD simulations was also consulted.

Summary:

This manuscript by the Accardi group proposes a new molecular mechanism to explain chloride and proton transport in CLC-class transporters and channels.

Using a combination of molecular dynamics, electrophysiology and non-canonical aminoacid substitutions, the authors find two phenylalanine residues that facilitate the movement of a protonated external glutamate involved both in gating and chloride permeation. This conformational change is different from the previously proposed one where the external glutamate occupies a chloride binding site and competes with the ion. This is an original and exciting paper and it constitutes an important contribution to understanding CLC channels/transporters.

Essential revisions:

1) In Figure 2—figure supplement 1F, the authors report PMF calculations that show that the preferred state of Phe357 for the Scen-Sext configuration is the 'up' rotamer. This is the rotamer observed in multiple crystal structures, e.g. ion-bound WT, E148Q, E148A EcCLC, WT apo EcCLC and WT CmCLC. Accordingly, one can see in Figure 2—figure supplement 1AB that the umbrella-sampling simulations that underlie the PMF in Figure 2A probe this 'up' rotamer when Z1 ~ 0 and Z2 ~ 5 A, i.e. the Scen-Sext configuration.

The reason why the Scen-Sext configuration is not a metastable state in the PMF in Figure 2A cannot be, therefore, that Phe357 is in the 'wrong' rotamer; to the contrary, the PMF in Figure 2A samples the 'correct' rotamer for Scen-Sext, according to the data provided in Figure 2—figure supplement 1AB. The reason must be instead that other configurations, and in particular those with Z1 < 0 and Phe357 'down', are more favored energetically when examined in simulation. Consistent with this, when Phe357 is artificially fixed in place in the 'up' rotamer (Figure 2D), the states with Z1 < 0 are strongly penalized.

It would seem that the central and clear prediction from this simulation data is that Scen-Sext with Phe357 up is an energy barrier for the mechanism of ion motion through EcClC, at 4-6 kcal/mol, and is strongly disfavored relative to e.g. Sint-Sext with Phe357 down (Figure 2-figure supplement 1B). Yet, the latter state is not included in the mechanistic diagram shown in Figure 8, which instead depicts the former as an intermediate (states V and VI), despite the PMF in Figure 2A. This is not internally consistent. More importantly, isn't this result/prediction counter to the E148Q structure (Phe357 up) and the analysis of relative ion occupancies therein (as well as in WT and E148A) by Lobet and Dutzler 2005? Also, isn't it at odds with the interpretation of the ITC experiments in Picollo et al., 2012 ("Our data suggest that protonation of Glu148 is unfavorable with no ions in the transport pathway, that protonation of the gating glutamate is favored by binding of a Cl− ion to Scen and is greatly stabilized by a second Cl− in Sex"). The authors should clarify if Scen-Sext with Phe357 up is a true free-energy minimum. The PMF in Figure 2A shows that Sint-Scen is not a metastable state either. Like Scen-Sext, it falls in the 4-6 kcal range, and is in a downhill gradient in free energy, which seems to suggest the ion in Sint isn't stable. This could be explained by the lower affinity of this site, but at 150 mM KCl, the simulation is well above saturating conditions.

A PMF calculation analogous to Figure 2A for E148Q with ions occupying Sext and Scen would help to alleviate these concerns – assuming it shows that the experimentally determined structure is a stable configuration.

2) The authors should provide an explanation of what factors cause Phe357 to favor a different rotamer when the ion configuration changes from Scen-Sext to Sint-Sext (Figure 2—figure supplement 1B) and from Scen-Sext to Sext only (Figure 2—figure supplement 1D), but somehow not from Scen-Sext to Scen only (Figure 2—figure supplement 1E). It is difficult to envisage how the sidechain of Phe357, which projects away from the chloride pathway and does not directly interact with the ions, nevertheless responds to the occupancy of the binding sites. Is this due to changes in the conformation of the structure?

3) What ion configuration was equilibrated for 100 ns before the PMF in Figure 2A was calculated? Were other starting conditions tested? How were the PMF calculations shown in Figure 2—figure supplement 1C-F initiated?

4) In regard to the proposed dipole-pi interaction between protonated Glu148 and the aromatic rings of Phe190 and Phe357: what is the magnitude of this interaction, as represented by the simulation forcefield, compared with e.g. a hydrogen-bond, all other factors being equal? Are the snapshots in Figure 3 anecdotal observations, or do they reflect statistically representative states? If the latter, how were they selected?

5) Mutants of Fext in CLC-5 (F255A) and CLC-7 (F301A) show rather voltage-independent currents, including steady state inward currents. There is a suspicion that they may be contaminated significantly by endogenous leak currents. Therefore, the authors have to provide positive evidence that these (inward) currents are real. Otherwise these data should be removed.

6) The fact that Gluex in CLC-1 structures seems to be in a position out of the Cl permeation pathway could be due to radiation damage in cryoEM data acquisition, to which glutamate residues are particularly sensitive. This should be discussed.

7) In a paper from 2003 (in which one of the lead authors of this manuscript is a co-author as well), Fcen of CLC-1 has been quite extensively characterized. For example, in addition to dramatic effects on gating, the F484A mutant showed a reduced single channel conductance. Also, in more recent papers from the Desaphy group, mutants of Fcen that cause myotonia have been characterized. This work should be discussed.

8) In the same 2003 paper (Estevez et al., 2003), Fcen was proposed to be part of an inhibitor binding site. This, together with recent structural data on the CLC-1 channel, should be discussed (e.g. could binding of these inhibitors impede the movement of Gluex?).

9) The outward movement of the protonated Gluex without interference somehow reminds me of a kinetic mechanism proposed for CLC antiporters (Jentsch and Pusch, 2018), in which a hypothetical "swap" between protonated Gluex and a Cl ion was proposed. This might be discussed.

10) Regarding the Sext Cl site, the recent paper from Park et al., from the Lim group should be discussed.

11) The experimental tests of the role of F190 and F357 in the CLC mechanism are not convincing. In CLC-ec1, the authors show that mutations at these positions decrease turnover rates and increase Cl^-^/H^+^ coupling stoichiometry. While this result can be called "consistent" with the MD simulations, it would also be consistent with any number of other mechanisms. Moreover, it is likely that mutating any other similarly conserved residue (for example F199) would cause similar changes in function. So, it is not clear that these results validate predictions of the MD simulations. Similarly, mutations on CLC-7 and CLC-0 could have many different interpretations. Unfortunately, it is not obvious what experimental tests could be done to specifically test predictions of the MD simulations. One experiment that would help would be to put F190 and F357 mutations into the CLC transporters in an uncoupled (E148A in CLC-ec1) background. Since the Phe residues are predicted to predominantly affect the H^+^-transport branch of the mechanism, they should have little or no effect (on Cl^-^ transport) in such a background. Ideally, it would be best if the authors could think of additional specific tests of predictions. If that is not feasible, it is important to remove a lot of overstatements that have been made with respect to interpretations of the experimental results. For example, most of the effects of the mutants are not "drastic" or "extreme" but are actually rather modest effects on function and much smaller than effects of other known CLC mutations. Also, the concluding sentence (…"our results show that the aromatic slide forms an evolutionarily conserved structural motif that is…") would need to be toned down.

12) The reduced current levels for the F301A and F514A mutants cannot be distinctly ascribed to the reduced transport rate observed in CLC-ec1. The effect of the mutant could also be a reduced expression level. The voltage-clamp experiments do not distinguish between these possibilities. The authors should provide evidence for similar expression levels or tone down this assertion.

13) The effects of the alanine substitutions in both external and central phenylalanines seem to be an enhancement of gating (current measurable at every voltage), without too drastic changes in voltage-dependence (z for WT and F514A is the same). Alternatively, could the inward current in the mutants be carried by protons? This would be a similar behavior to omega currents in the voltage sensor domain proteins. This distinction is important because the effects of atomic mutagenesis are divergent with those of alanine substitution. Substitutions of the central phenylalanine make gating more difficult, while alanine substitutions seem to enhance gating.

14) It would have been nice if non- canonical aminoacid incorporation experiments were carried out in the CLC-ec1 transporter, for which the MD simulations were performed. Is there a reason for this? Although it seems that the effects are conserved, it seems also that gating and permeation effects of mutagenesis are mixed in the CLC channels, while transport could be better characterized in CLC-ec1.

---

## [Author Response]

Essential revisions:1) In Figure 2—figure supplement 1F, the authors report PMF calculations that show that the preferred state of Phe357 for the Scen-Sext configuration is the 'up' rotamer. This is the rotamer observed in multiple crystal structures, e.g. ion-bound WT, E148Q, E148A EcCLC, WT apo EcCLC and WT CmCLC. Accordingly, one can see in Figure 2—figure supplement 1AB that the umbrella-sampling simulations that underlie the PMF in Figure 2A probe this 'up' rotamer when Z1 ~ 0 and Z2 ~ 5 A, i.e. the Scen-Sext configuration.

We thank the reviewer for raising this point. First, we would like to emphasize that in our PMF calculation shown in Figure 2A, F357 visits both the down and up states. Thus, it is not in the “right” or “wrong” conformer, but samples both. This illustrates that both conformers of Phe357 are accessible on the time scale of the simulation. However, the plots of Figure 2—figure supplement 1AB reveal that χ1(F357) is a slow degree of freedom, i.e. that the transition along this reaction coordinate is not fast enough to be thoroughly sampled in absence of a biasing potential, especially when ions enter or exit the Scen and Sext binding sites. Thus, special care must be taken in interpreting the sampling of the conformations of F357. In theory, this could be investigated with an exhaustive PMF calculation that would involve three reaction coordinates: the position of the two Cl^-^ ions and the χ1(F357) angle. However, such 3D PMF is computationally expensive, not strictly required and likely insufficient as other factors such as the influence of the protonation state of E148 on the interactions with the aromatic ring of F357 or that of conformational rearrangements (such as those recently reported by Chavan et al., 2019) would also need to be considered. Such an expansive set of additional calculations, while interesting, is beyond the scope of the present manuscript. The reason we believe the 3D calculation is not strictly required is that our present set of calculations identifies the 2 key degrees of freedom of the system: (1) how the conformation of F357 affects ion occupancy of the CLC pathway (Figure 2), and (2) how ion occupancy influences the rotameric arrangement of F357 (Figure 2—figure supplement 1). Additionally, we validate these inferences by showing that an inhibitory crosslink prevents the rearrangement of F357 (Figure 2—figure supplement 2). These 1D and 2D PMFs correspond to key degrees of freedom in the 3D PMF discussed above. This point is now discussed in subsection “Rotation of F357 enables the formation of an aromatic slide”.

The reason why the Scen-Sext configuration is not a metastable state in the PMF in Figure 2A cannot be, therefore, that Phe357 is in the 'wrong' rotamer; to the contrary, the PMF in Figure 2A samples the 'correct' rotamer for Scen-Sext, according to the data provided in Figure 2—figure supplement 1AB. The reason must be instead that other configurations, and in particular those with Z1 < 0 and Phe357 'down', are more favored energetically when examined in simulation. Consistent with this, when Phe357 is artificially fixed in place in the 'up' rotamer (Figure 2D), the states with Z1 < 0 are strongly penalized.

It is true that the Phe357 conformers favor different ion configurations. Figure 2C-D reveal that Phe357 ‘up’ stabilizes the 2-ion state S_cen_- S_ext_ without excessively destabilizing other states like S_cen_-S_ext_* or S_int_-S_ext_, while Phe357 ‘down’ destabilizes the S_cen_-S_ext_ configuration. The opposite effect of the ‘up’ and ‘down’ conformers on the ion occupancy state is central to the regulation of ion transport. In a transporter, at each cycle, the transition from the ‘up’ to the ‘down’ conformer would favor the release of the ions and the proton. The PMFs of Figure 2C-D are compatible with this view, although the ‘up’ conformer slightly destabilizes some ion configurations.

It would seem that the central and clear prediction from this simulation data is that Scen-Sext with Phe357 up is an energy barrier for the mechanism of ion motion through EcClC, at 4-6 kcal/mol, and is strongly disfavored relative to e.g. Sint-Sext with Phe357 down (Figure 2—figure supplement 1B). Yet, the latter state is not included in the mechanistic diagram shown in Figure 8, which instead depicts the former as an intermediate (states V and VI), despite the PMF in Figure 2A. This is not internally consistent.

We thank the reviewer for pointing this out. However, the binding affinity to S_int_ is low as there is no free energy barrier preventing the release of the ion, which is in equilibrium between the binding site and the bulk solution. Accordingly, an ion could bind to S_int_ from the intracellular milieu in states IV-V-VI-VII, but as long as the intracellular gate remains closed it would not participate in the permeation process. In the counterclockwise transition from state (VI) to (V), an ion enters from the extra-cellular side (not the internal side), and thus the ion S_int_ would be released without ever reaching S_cen_. This is now discussed in subsection “Formation of an aromatic slide is essential to Glu_ex_ movement”.

More importantly, isn't this result/prediction counter to the E148Q structure (Phe357 up) and the analysis of relative ion occupancies therein (as well as in WT and E148A) by Lobet and Dutzler 2005? Also, isn't it at odds with the interpretation of the ITC experiments in Picollo et al., 2012 ("Our data suggest that protonation of Glu148 is unfavorable with no ions in the transport pathway, that protonation of the gating glutamate is favored by binding of a Cl− ion to Scen and is greatly stabilized by a second Cl− in Sex"). The authors should clarify if Scen-Sext with Phe357 up is a true free-energy minimum.

As described above, the conformation with S_cen_- S_ext_ occupied and with Phe357 ‘up’ is a local free-energy minimum and not a barrier. Thus, our PMF calculations presented in Figure 2 are coherent with the analysis by Lobet and Duztler, 2005 and Picollo et al., 2012.

The PMF in Figure 2A shows that Sint-Scen is not a metastable state either. Like Scen-Sext, it falls in the 4-6 kcal range, and is in a downhill gradient in free energy which seems to suggest the ion in Sint isn't stable. This could be explained by the lower affinity of this site, but at 150 mM KCl, the simulation is well above saturating conditions.

The affinity of the S_int_ site for Cl^-^ is indeed low, with best estimates placing it at a K_d_ >20 mM (Picollo et al., 2009). Indeed, at 250 mM Br^-^ the anomalous electron density of the S_int_ site is ~88% of that of S_cen_, suggesting that even at this very high concentration S_int_ might not be fully saturated (Accardi et al., 2006). Thus, our results are consistent with the data in the literature. However, PMF calculations describe the energetics of interaction between the restrained ions and the protein, but not the equilibrium between ions in the bulk and those in the pore (there is no exchange between the two ensembles). The ion concentration in the PMF calculations is set so that the overall energetics is as close to experimental conditions as possible, though the PMF is mostly independent of the ion concentration. Addressing the question of the saturation of a binding site at a given concentration would require Brownian dynamics simulations on an energy surface represented by the PMF (e.g. Bernèche and Roux, 2003), which, while interesting, are outside the scope of our current work.

A PMF calculation analogous to Figure 2A for E148Q with ions occupying Sext and Scen would help to alleviate these concerns – assuming it shows that the experimentally determined structure is a stable configuration.

We thank the reviewers for the helpful suggestion. However, we note that the structure used for the PMF of Figure 2A is equivalent to that of the E148Q structure: the E148 side chain is protonated, and out of the pathway in a conformation modeled on that of the E148Q crystal structure. Starting the PMF calculation with ions in S_cen_-S_ext_ with Phe357 in the ‘up’ state would show a more stable S_cen_-S_ext_ configuration simply because the sampling of the Phe357 ‘up’ state would be increased. However, the calculation would not necessarily converge because, as discussed above, χ1(F357) fluctuates between two conformers. To avoid any uncertainty on the conformational sampling, the best approach is to control the χ1(F357) with a harmonic restraint, as presented in Figure 2B and 2C.

2) The authors should provide an explanation of what factors cause Phe357 to favor a different rotamer when the ion configuration changes from Scen-Sext to Sint-Sext (Figure 2—figure supplement 1B) and from Scen-Sext to Sext only (Figure 2—figure supplement 1D), but somehow not from Scen-Sext to Scen only (Figure 2—figure supplement 1E). It is difficult to envisage how the sidechain of Phe357, which projects away from the chloride pathway and does not directly interact with the ions, nevertheless responds to the occupancy of the binding sites. Is this due to changes in the conformation of the structure?

The key element is whether Scen is occupied by an ion or not: When only one ion is present in either S_cen_ or S_ext_, the backbone NH group of Phe357 reorient itself to interact with that ion. When both S_cen_ and S_ext_ are occupied, the NH group of Phe357 interacts mostly with the ion in S_cen_. The same applied to the neighboring Ile356 residue. Thus, the ion occupancy state of S_cen_ and S_ext_ impacts on the backbone conformation of residues Ile356 and Phe357, which at its turn modifies the free energy landscape of the F357 side chain. This is now discussed in the legend to Figure 2—figure supplement 1.

3) What ion configuration was equilibrated for 100 ns before the PMF in Figure 2A was calculated? Were other starting conditions tested? How were the PMF calculations shown in Figure 2—figure supplement 1C-F initiated?

The starting configuration for the PMF of Figure 2A involves ions in sites S_cen_ and S_ext_. Other starting conditions were tested but typically led to bigger hysteresis (free energy difference between the S_int_/S_ext_ and S_ext_/S_ext_^*^ states). The PMF of Figure 2—figure supplement 1 were initiated with ions at positions specified in the different panels and with the F357 side chain in the down state (-70º). We have repeated the calculations starting in the up state (-160º), which led qualitatively to the same results. We have replotted the PMF showing both calculations. This is now detailed in the legend to Figure 2—figure supplement 1 and Figure 2—figure supplement 2.

4) In regard to the proposed dipole-pi interaction between protonated Glu148 and the aromatic rings of Phe190 and Phe357: what is the magnitude of this interaction, as represented by the simulation forcefield, compared with e.g. a hydrogen-bond, all other factors being equal?

Using the charmm36 force field, we calculated that the interaction between a protonated glutamate side chain and the aromatic ring of a phenylalanine is -5.0 kcal/mol (for an optimized conformation in vacuum). For comparison, the interaction between a protonated glutamate and a serine side chain is -5.8 and -7.2 kcal/mol, depending if the unprotonated or the protonated oxygen of the carboxylate group is involved. These number are in line with ab initio calculations by Du et al., (2013), who reported interactions on the order of -2.4 to -7.1 kcal/mol. This is now added to the Materials and methods section.

Are the snapshots in Figure 3 anecdotal observations, or do they reflect statistically representative states? If the latter, how were they selected?

The snapshots of Figure 3 appear spontaneously in PMF calculations for which the positions of ions were controlled. The illustrated side chain and water wire conformations are observed in at least 3 sampling windows and remain stable for most of the 500 ps sampling time of a given window. This is now mentioned in the legend to Figure 3.

5) Mutants of Fext in CLC-5 (F255A) and CLC-7 (F301A) show rather voltage-independent currents, including steady state inward currents. There is a suspicion that they may be contaminated significantly by endogenous leak currents. Therefore, the authors have to provide positive evidence that these (inward) currents are real. Otherwise these data should be removed.

We thank the reviewers for raising this important concern. Several lines of evidence suggest that the observed currents are indeed mediated by the expressed mutant proteins:

i) In oocytes batches where we expressed WT and mutant CLC-5 and CLC-7 constructs, no significant inward currents could be detected in oocytes injected with the WT constructs. As an example, Author response image 1 shows average I-V relationships from multiple oocytes from independent batches for CLC-7 WT and F301A and from 1 batch for CLC-5 WT and F255A.

ii) If the currents measured in oocytes expressing CLC-7 F301A and CLC-5 F255A were contaminated by linear leak currents, then we would expect that (i) the shape of the I-V for these mutants should be more variable than that of the WT counterpart, and (ii) the rectification index R=I(+80mV)/I(-80mV) should vary in these mutants as a function of the total current. Neither is the case. Figure 5E, Figure 5—figure supplement 2D show that the errors in the normalized I-V relationships for the WT and mutant CLC-7 and CLC-5 constructs are comparable. Further, there is no clear correlation between the rectification index R and the value of the current measured at -80 mV, which is where the leak should be most significant (Author response image 1).

iii) We measured how CLC-7 F301A depends on [Cl^-^]_ext_ and [H^+^]_ext_. We reasoned that if the effects of the F301A mutation in CLC-7 mirrored those seen in CLC-ec1 then we would expect that the V_rev_ of the currents should depend on Cl^-^, and only minimally on H^+^, due to the degraded exchange stoichiometry. This is indeed the case: when [Cl^-^]_ext_ is reduced from ~103 mM to ~17 mM we observe a reduction of the current at positive voltages, an increase of the current at negative voltages and a right shift of V_rev_, consistent with the idea that Cl^-^ is the main charge carrier of the observed currents. Conversely, a 10-fold decrease in [H^+^]_ext_ elicits only a ~5-10 mV left-shift of V_rev_ of the CLC-7 F301A currents and a ~10% increase in the steady state current at +90 mV (as opposed to the ~30-40% increase in current seen in the WT protein). These findings are consistent with the idea that this mutant degrades H^+^/Cl^-^ coupling. The inherent difficulties in controlling the intracellular [Cl^-^] and [H^+^] concentration in oocytes prevent us from precisely determining the exchange stoichiometry of the F301A mutant, but these results are in qualitative agreement with the effects seen with CLC-ec1 F190A. This additional data is now shown in Figure 5—figure supplement 1A, B and discussed in subsection “The role of Pheex and Phecen is conserved in mammalian transporters” of the revised manuscript.

Lastly, we note that the effects of the Phe_ex_ mutants are qualitatively conserved across CLC-7, CLC-5 and CLC-0, as in all three homologues the voltage dependence of the currents at negative voltages is drastically reduced (Figure 5B,E,F; Figure 5—figure supplement 2B,D; Figure 6B,D). This is now mentioned in subsection “Pheex and Phecen are important for CLC channel gating”.

**Author response image 1. respfig1:** Inward currents elicited by CLC-7 and CLC-5 Phe_ex_ mutants are not due to conductances endogenous to the expression system. Comparison of normalized I-V relationships of CLC-7 WT and F301A (A) as well as of CLC-5 WT and F255A (B) are shown for individual, independent batches of *X. laevis* oocytes. Endogenous conductances are assumed to be constant between WT and mutant injected oocytes within the same batch. No prominent inward currents were present in WT expressing oocytes, indicating that the inward currents are likely to be specific to Phe_ex_ mutants (CLC-7: N(WT, Batch 1)=5, N(F301A, Batch 1)=3, N(WT, Batch 2)=2, N(F301A, Batch 2)=2; CLC-5: N(WT)=2, N(F255A)=5). N refers to the number of individual oocytes. All values are reported as mean ± S.E.M, error bars are not shown where they are smaller than the symbol size. (C, D) For CLC-7 F301A (C; N=19) and CLC-5 F255A (D; N=10) no correlation was observed between the rectification index R=I_+80mV_/I_-80mV_ and the current measured at -80 mV.

6) The fact that Gluex in CLC-1 structures seems to be in a position out of the Cl permeation pathway could be due to radiation damage in cryoEM data acquisition, to which glutamate residues are particularly sensitive. This should be discussed.

We thank the reviewers for raising this point. However, in the CLC-1 map (EMD-7544) the density corresponding to E232 is well-defined (Author response image 2), so we feel confident in leaving our statement in without additional disclaimers.

**Author response image 2. respfig2:** Close up view of the density map (gray) in the CLC-1 pore allows placement of the E232 side chain (cyan, ball and stick).

7) In a paper from 2003 (in which one of the lead authors of this manuscript is a co-author as well), Fcen of CLC-1 has been quite extensively characterized. For example, in addition to dramatic effects on gating, the F484A mutant showed a reduced single channel conductance. Also, in more recent papers from the Desaphy group, mutants of Fcen that cause myotonia have been characterized. This work should be discussed.8) In the same 2003 paper (Estevez et al., 2003), Fcen was proposed to be part of an inhibitor binding site. This, together with recent structural data on the CLC-1 channel, should be discussed (e.g. could binding of these inhibitors impede the movement of Gluex?).

We thank the reviewers for pointing this out. The results of the Estevez et al., and of the Imbrici et al., papers are now discussed in subsection “A mechanism for CI^-^/H^+^ exchange”.

9) The outward movement of the protonated Gluex without interference somehow reminds me of a kinetic mechanism proposed for CLC antiporters (Jentsch and Pusch, 2018), in which a hypothetical "swap" between protonated Gluex and a Cl ion was proposed. This might be discussed.

We thank the reviewers for pointing out the similarities between our proposed mechanism and that suggested in the Jentsch and Pusch review. This is now discussed in subsection “A mechanism for CI^-^/H^+^ exchange”.

10) Regarding the Sext Cl site, the recent paper from Park et al., from the Lim group should be discussed.

We now discuss the recent Park et al., paper in the Introduction.

11) The experimental tests of the role of F190 and F357 in the CLC mechanism are not convincing. In CLC-ec1, the authors show that mutations at these positions decrease turnover rates and increase Cl^-^/H^+^ coupling stoichiometry. While this result can be called "consistent" with the MD simulations, it would also be consistent with any number of other mechanisms.

We agree with the reviewers that the effects of each individual mutant by itself does not validate the predictions of the MD simulations. However, the results of multiple mutants across 4 different CLC proteins show comparable effects. Moreover, the incorporation of the rotationally restricted Phe derivative 2,6diMeth-Phe at Phe_cen_ of CLC-0 impaired the opening of the single-pore gate (Figure 7C), providing a direct test of the MD predictions that the rotational movement of Phe_cen_ facilitates the movement of Gluex in and out of the Cl^-^ permeation pathway. This point is now clarified in subsection “Pheex and Phecen are important for CLC channel gating”.

Moreover, it is likely that mutating any other similarly conserved residue (for example F199) would cause similar changes in function. So, it is not clear that these results validate predictions of the MD simulations.

We thank the reviewers for raising this point. We did test the effects of the F223A mutant of CLC-0, which corresponds to F199 in CLC-ec1. The single-pore gate G-V of the F223A mutant is slightly right-shifted compared to that of the WT channel, the Boltzmann function parameters are: V_0.5_ = -47±3 mV, z = 0.7±0.03, *P*_min_ = 0.02±0.02 (data not shown). Thus, while Ala substitutions at Phe_ex_ and Phe_cen_ keep the fast gate open, the F223A mutation promotes pore closure, supporting the specificity of the effects of Phe_ex_ and Phe_cen_. The F223A mutation has a drastic effect on the common pore gate, consistent with its location near the dimer interface. These results are beyond the scope of the present manuscript and thus we would prefer not to include them.

Similarly, mutations on CLC-7 and CLC-0 could have many different interpretations. Unfortunately, it is not obvious what experimental tests could be done to specifically test predictions of the MD simulations. One experiment that would help would be to put F190 and F357 mutations into the CLC transporters in an uncoupled (E148A in CLC-ec1) background. Since the Phe residues are predicted to predominantly affect the H^+^-transport branch of the mechanism, they should have little or no effect (on Cl^-^ transport) in such a background.

We thank the reviewers for this suggestion. We have now tested the effects of the F190 and F357 mutants in the background of the E148A mutant of CLC-ec1. As the reviewers predicted, the F357A/E148A mutant has the same Cl^-^ transport rate as the single E148A mutant (Figure 4E-F). Remarkably, the double F190A/E148A mutant “rescues” the transport defect of the parent single mutants and has WT-like transport rates. The non-additivity of the effects of the F357A and F190A mutations with that of the E148A mutant qualitatively supports the idea that these residues interact during transport. This new data is now shown in Figure 4 and discussed in subsection “The aromatic slide residues are essential for Cl^-^:H^+^ 216 coupling and exchange in CLC-ec1”.

Ideally, it would be best if the authors could think of additional specific tests of predictions. If that is not feasible, it is important to remove a lot of overstatements that have been made with respect to interpretations of the experimental results. For example, most of the effects of the mutants are not "drastic" or "extreme" but are actually rather modest effects on function and much smaller than effects of other known CLC mutations. Also, the concluding sentence (…"our results show that the aromatic slide forms an evolutionarily conserved structural motif that is…") would need to be toned down.

We agree with the reviewers that our results suggest that the aromatic slide plays a key role in transport, rather than demonstrate such a role. We have toned down our conclusions throughout the manuscript and provided additional experimental evidence supporting our mechanistic inferences.

12) The reduced current levels for the F301A and F514A mutants cannot be distinctly ascribed to the reduced transport rate observed in CLC-ec1. The effect of the mutant could also be a reduced expression level. The voltage-clamp experiments do not distinguish between these possibilities. The authors should provide evidence for similar expression levels or tone down this assertion.

We thank the reviewers for pointing this out. We have removed this statement from the manuscript.

13) The effects of the alanine substitutions in both external and central phenylalanines seem to be an enhancement of gating (current measurable at every voltage), without too drastic changes in voltage-dependence (z for WT and F514A is the same). Alternatively, could the inward current in the mutants be carried by protons? This would be a similar behavior to omega currents in the voltage sensor domain proteins.

We thank the reviewers for bringing up this interesting hypothesis. We now show that the F301A mutant impairs H^+^ transport by CLC-7 and the majority of the current is carried by Cl^-^ (Figure 5—figure supplement 1B), consistent with our results on CLC-ec1. The F514A mutant on the other hand has minimal inward current and therefore could not be tested.

This distinction is important because the effects of atomic mutagenesis are divergent with those of alanine substitution. Substitutions of the central phenylalanine make gating more difficult, while alanine substitutions seem to enhance gating.

We note that while two unnatural amino acid substitutions at Phe_cen_ make channel opening more difficult by inducing a right shift in the G-V (Cha and 2,6diMeth-Phe) the third (2,6F_2_-Phe) induces a left shift in G-V and thus makes channel opening easier, in the same direction as the Ala substitution (Figure 7C). A similar pattern holds true for the effects of the substitutions on the common-pore gate (Figure 7E). Thus, we do not agree that there is a general divergence between the effects of the canonical and non-canonical substitutions. The differential effects of the ncAA substitutions on gating reflect the different nature of the introduced manipulations and highlight the ability of this approach to perturb specific interactions.

14) It would have been nice if non- canonical aminoacid incorporation experiments were carried out in the CLC-ec1 transporter, for which the MD simulations were performed. Is there a reason for this? Although it seems that the effects are conserved, it seems also that gating and permeation effects of mutagenesis are mixed in the CLC channels, while transport could be better characterized in CLC-ec1.

We thank the reviewers for this interesting suggestion. Unfortunately, there are no available tRNA/aminoacyl-tRNA-synthetase pairs for prokaryotic expression systems that enable the specific incorporation of the Phe derivatives used here, thus preventing us from carrying out these experiments.